# Design and Analysis of an Autonomous Active Ankle–Foot Prosthesis with 2-DoF

**DOI:** 10.3390/s25164881

**Published:** 2025-08-08

**Authors:** Sayat Akhmejanov, Nursultan Zhetenbayev, Aidos Sultan, Algazy Zhauyt, Yerkebulan Nurgizat, Kassymbek Ozhikenov, Abu-Alim Ayazbay, Arman Uzbekbayev

**Affiliations:** 1Department of Robotics and Technical Tools of Automation, Satbayev University, Almaty 050013, Kazakhstan; akhmejanov.s@stud.satbayev.university (S.A.); a.sultan@aues.kz (A.S.); k.ozhikenov@satbayev.university (K.O.);; 2Department of Electronic Engineering, Almaty University of Power Engineering and Telecommunications, Almaty 050013, Kazakhstan; a.zhauyt@aues.kz; 3Department of Automation and Control, Mukhametzhan Tynyshbayev ALT University, Almaty 050013, Kazakhstan

**Keywords:** active prosthesis, ankle joint, two degrees of freedom (2-DoF), stepper motor, ball screw mechanism

## Abstract

This paper presents the development, modeling, and analysis of an autonomous active ankle prosthesis with two degrees of freedom (2-DoF), designed to reproduce movements in the sagittal (dorsiflexion/plantarflexion) and frontal (inversion/eversion) planes in order to enhance the stability and naturalness of the user’s gait. Unlike most commercial prostheses, which typically feature only one active degree of freedom, the proposed device combines a lightweight mechanical design, a screw drive with a stepper motor, and a microcontroller-based control system. The prototype was developed using CAD modeling in SolidWorks 2024, followed by dynamic modeling and finite element analysis (FEA). The simulation results confirmed the achievement of physiological angular ranges of ±20–22 deg. in both planes, with stable kinematic behavior and minimal vertical displacements. According to the FEA data, the maximum von Mises stress (1.49 × 10^8^ N/m^2^) and deformation values remained within elastic limits under typical loading conditions, though cyclic fatigue and impact energy absorption were not experimentally validated and are planned for future work. The safety factor was estimated at ~3.3, indicating structural robustness. While sensor feedback and motor dynamics were idealized in the simulation, future work will address real-time uncertainties such as sensor noise and ground contact variability. The developed design enables precise, energy-efficient, and adaptive motion control, with an estimated average power consumption in the range of 7–9 W and an operational runtime exceeding 3 h per charge using a standard 18,650 cell pack. These results highlight the system’s potential for real-world locomotion on uneven surfaces. This research contributes to the advancement of affordable and functionally autonomous prostheses for individuals with transtibial amputation.

## 1. Introduction

Over the past decade, significant progress has been made in the field of bionic prosthetics. These devices are robotic systems based on microcontrollers, enabling the restoration of biological functions beyond the capabilities of traditional solutions [1]. Unlike passive prostheses, modern active ankle devices provide dynamic motion control, which positively affects gait efficiency in individuals with lower-leg amputations [2]. This progress highlights a major shift toward more functional designs aimed at mimicking natural biomechanics.

Despite the advances in active prosthesis development, all commercially available devices, including the Ottobock Empower [3] and Össur Proprio [4], offer only a single degree of freedom (1-DoF) in the sagittal plane. They are limited to plantar flexion and dorsiflexion movements, whereas studies show that effective balance control also requires motion in the frontal plane (inversion and eversion).

Several attempts have been made to develop ankle prostheses with two degrees of freedom. In [5], a concept of an active ankle–foot prosthesis with inversion and eversion functionality is presented, aimed at imitating natural limb movement and enhancing user stability. The device combines active and passive control elements, enabling adaptation to uneven terrain, reducing energy expenditure during walking, and dampening impact loads. However, its primary drawbacks include the need for external power, the complexity of biosignal processing, and limited active control in the frontal plane. Moreover, the device exists only as a 3D model, with no clinical validation of its effectiveness.

Another study [6] investigates functional testing of the passive MyFlex-δ foot prosthesis, which utilizes energy storing and return (ESR) technology. Its spherical ankle joint allows adaptation to uneven surfaces and improves gait stability. However, the lack of active control, the limited range of motion, and potential energy losses during load transfer remain major limitations. Study [7] presents an active ankle–foot prosthesis capable of replicating the biomechanics of a natural joint and reducing metabolic cost by 14% compared with passive devices. However, the system is heavy, requires individual calibration, and is battery-dependent. Work by [8] proposes an optimized prosthesis with a series elastic actuator (SEA) that reduces peak energy consumption. This approach lowered peak motor power by 78.8% at a speed of 1.1 m/s and by 66.98% at 1.6 m/s. Nevertheless, the design’s complexity and calibration demand remain limiting factors. Study [9] introduces a bionic lower limb prosthesis with a neuro-controlled system. The use of 3D printing reduces the cost and weight. However, testing revealed system response delays, and the strength of PLA materials remains questionable. Study [10] presents the concept of a 2-DoF active ankle prosthesis that supports inversion and eversion. While the prosthesis demonstrates a high control accuracy, the drawbacks include structural complexity, increased weight, and limited motion ranges. Research by [11] focuses on optimizing resistance torque control, enhancing the biomechanical imitation of a natural ankle joint. Using a five-link spring mechanism, the torque transmission error was reduced by 66.46%. However, the control system still requires further optimization. Work by [12] proposes an active transtibial prosthesis using a seven-link inverse dynamics model of gait. This allows real-time motion prediction but results in increased metabolic cost and complex calibration requirements. Study [13] describes an active ankle prosthesis controlled via a neuromuscular model. The control system adapts to terrain changes but is sensitive to latency and requires frequent recharging. Work by [14] introduces a 2-DoF active ankle prosthesis using Bowden cables for motion control in both planes. Brushless motors reduce the metabolic cost, but the energy dependency remains unsolved. Study [15] examines the development of a soft robotic ankle prosthesis with flexible pneumatic actuators. This reduces impact loads but requires a compressed air source and restricts motion range. Study [16] presents a flexible bionic ankle prosthesis made from soft polyurethane and carbon fiber. The device improves load distribution but remains passive, limiting the functionality compared with active models. Work by [17] focuses on integrating hip, knee, ankle, and foot prostheses into a unified system with a motorized hip joint. However, testing was conducted on a small sample size, requiring further research. In [18], a transfemoral prosthesis with a magnetorheological (MR) damper is developed, enhancing stability but incurring high production costs and requiring energy optimization. Study [19] is dedicated to developing a reactive series elastic actuator (RFSEA), which improves gait smoothness and reduces user load. However, it requires external power and complex personalized calibration.

To complement the qualitative literature review, a comparative table was compiled based on the data extracted from the most relevant studies [5,6,7,8,9,10,11,12,13,14,15,16,17,18,19]. Table 1 presents key quantitative metrics such as device mass, torque output, range of motion, and power consumption for existing 2-DoF ankle–foot prosthesis prototypes, providing a clearer context for assessing the novelty and advantages of the proposed design.

Despite extensive research and development, there is still not a fully autonomous 2-DoF active ankle prosthesis that

–Has a compact and lightweight design without excessive mass;–Provides active control in both the sagittal and frontal planes;–Features low power consumption and high autonomy;–Is validated through large-scale clinical trials.

The development of such a prosthesis would significantly improve users’ quality of life by ensuring natural motion, stability, and walking comfort, while expanding mobility options under challenging conditions.

The present work is aimed at developing an autonomous active ankle prosthesis with two degrees of freedom (2-DoF), providing active control of motion in both the sagittal and frontal planes.

The proposed system includes

–The proposed system includes a mechanical structure designed to balance reliability, low weight, and adaptability to gait dynamics;–Electronic systems and sensors ensuring precise motion control;–Intelligent control algorithms mimicking the biomechanics of natural gait;–Autonomous operation focused on minimizing power consumption and maximizing runtime without recharging.

This approach not only expands the functional capabilities of the device but also significantly improves the adaptability to various surface types, enhancing comfort and natural user motion.

The development of a 2-DoF active ankle prosthesis marks a significant step forward in robotic prosthetic technology. Unlike existing counterparts, the proposed device will enable active control of inversion/eversion and plantar/dorsiflexion, improving user balance, surface adaptation, and walking stability.

The system is designed to generate sufficient mechanical power for natural locomotion, reducing the load on other joints and supporting gait symmetry. Intelligent control algorithms will allow real-time movement analysis and adaptive responses based on the walking environment.

A high energy efficiency and autonomy will extend the operation time without frequent charging, improving daily usability. The implementation of such a prosthesis could significantly enhance mobility, comfort, and natural movement, facilitating user socialization and societal integration.

To achieve these goals, the study focuses on the following key questions:–How can we design a 2-Dof active ankle prosthesis structure that balances strength, weight, and functionality?–What kinematic and dynamic principles should guide the actuator system design for sagittal and frontal plane movements?–Which control algorithms best adapt prosthetic motion to terrain variations and user gait dynamics?–How can power consumption be minimized while maintaining a high output and system stability?–How does active ankle joint control affect gait symmetry, metabolic cost, and overall user comfort?

This paper provides a review of current developments, analyzing their strengths and limitations, and proposes a new concept of an autonomous powered prosthesis with two degrees of freedom. This study addresses the biomechanical requirements of an ankle prosthesis, including key motion parameters, torque, mechanical power, and stabilization needs during walking.

A kinematic mechanism for sagittal and frontal motion is developed. The design process is based on CAD modeling, followed by structural optimization, material selection, and actuator and sensor system analysis to ensure reliability and functionality.

The paper also presents the structural design of the initial prototype of the ankle prosthesis developed within the framework of the study. To evaluate the effectiveness of the proposed solution, modeling is performed, including design validation, kinematic, dynamic, and stability analysis based on numerical simulation and finite element method (FEM). The results confirm that the proposed active prosthesis concept closely mimics the biological function of the shank–foot system, supporting natural and comfortable locomotion.

In conclusion, this work aims to create an innovative 2-DoF active ankle prosthesis that brings artificial limb functionality closer to the natural capabilities of the biological shank–foot system. The proposed development has the potential to significantly enhance the quality of life for individuals with amputations and broaden their mobility in everyday settings.

## 2. Materials and Methods

This section briefly outlines the development stages and research methods for the active ankle prosthesis with two degrees of freedom. Based on the analysis of biomechanical requirements, an optimized CAD model was created, a kinematic scheme was implemented, and a functional prototype was fabricated using 3D printing. The physical prototype was fabricated using PLA plastic through FDM 3D printing for proof-of-concept evaluation. For the finite element analysis (FEA), the prosthesis structure was modeled using a AISI 304 stainless steel to assess the performance of the intended final design under typical loading conditions.

### 2.1. Biomechanical Requirements for Ankle–Foot Prosthesis

The ankle joint is a critically important component of lower limb biomechanics, providing support, shock absorption, stabilization, and energy transfer during walking. Its anatomical and functional complexity requires careful consideration of all biomechanical characteristics in the design of an active prosthesis capable of replicating natural movement.

The ankle joint performs movements in multiple planes. In the sagittal plane, it enables dorsiflexion of approximately 20° and plantarflexion in the range of 25–30°, allowing the foot to adapt to various phases of the gait cycle. In the frontal plane, inversion and eversion occur within ±10–15°, which are essential for compensating for lateral shifts and adapting to uneven surfaces (Figure 1).

During the full gait cycle, which includes the phases of heel strike, mid-stance, toe-off, and swing phase, the ankle joint performs the following key functions:–At heel strike, the joint absorbs impact forces and initiates plantarflexion. Dorsiflexion begins after the foot is flat and increases toward toe-off.–During mid-stance, it stabilizes the body over the supporting leg.–In the toe-off phase, it generates the necessary torque and energy to propel the body forward.–During the swing phase, it prepares the foot for the next step (Figure 2).

The torque generated by the ankle joint during the toe-off phase reaches 100–150 N/m, which is essential for the effective forward propulsion of the body. An active prosthesis must replicate this torque with the ability to adaptively modulate it based on walking speed and surface conditions. To achieve this, angular position and load sensors such as the BMI160 IMU are used, providing real-time feedback.

The joint’s energetic characteristics involve performing up to 0.2 J/kg of positive mechanical work per step. To mimic these parameters, the prosthesis employs energy-efficient actuators capable of storing and releasing energy in synchronization with the gait phases. Data processing and adaptive control are managed using the NVIDIA Jetson platform, which enables real-time sensor data analysis and the prediction of required forces.

Stability and surface adaptation are critical aspects, especially when walking on uneven or inclined terrain. The prosthesis must respond to changes in surface angle, obstacle height, and texture. This is achieved through a sensor system and adaptive control algorithms that generate feedback and adjust actuator behavior to maintain user balance (Figure 3).

Shock absorption and load distribution in the prosthesis replicate the function of the biological joint in reducing impact forces on the musculoskeletal system. The design incorporates spring elements, dampers, and energy-efficient materials, as well as active damping systems that adjust stiffness in real time based on the walking conditions. This ensures stump protection and enhances user comfort.

Gait cycle synchronization is achieved through sensory systems (IMU, encoders, pressure sensors) that accurately detect the phases of movement: heel strike, mid-stance, toe-off, and swing phase. Software solutions based on the NVIDIA Jetson platform process sensor data and adapted actuator performance to ensure smooth and efficient motion.

The response to external forces such as changes in speed or direction is enabled by an intelligent control system capable of predicting and adapting to dynamic environmental conditions. This provides the user with stability, safety, and freedom of movement.

Thus, an active ankle prosthesis must replicate not only the kinematics and dynamics of a healthy joint, but also deliver mechanical power, adapt to surface conditions, provide shock absorption, ensure stability, synchronize with the gait cycle, and respond to external forces. The key biomechanical parameters of the natural joint and the corresponding prosthetic requirements are presented in Table 2.

### 2.2. Kinematic Scheme

To analyze the motion of the ankle prosthesis, a kinematic model was developed to describe the rotational movement of the foot segment relative to a given axis. This model is based on coordinate transformation during rotation by an angle α. Such an approach allows for the accurate calculation of point positions in space during movement.

The kinematic model is divided into the following three main components:(a)The transformation of coordinates during rotation;(b)The determination of the positions of key points a, b, and c;(c)The calculation of the relationship between angular displacement and force/torque.

The formulas presented below describe the spatial motion that occurs during the movement phase of the exoskeleton.

The trajectory of motion between points A and B is shown. Point C represents the element applying the load to the exoskeleton, while M denotes the motor. This schematic illustrates the required motor force and geometric constraints necessary to follow the specified trajectory.

Figure 4 and Figure 5 jointly illustrate the motion generation process in the 2-DoF mechanism. In Figure 4, the motor (denoted as M) drives the crank positioned at point A, producing a circular motion. This rotational motion is transferred through the connecting rod to point C, which changes its position as the crank rotates. In Figure 5, the successive positions of point C are shown for three configurations, A1, A2, and A3, representing different phases of ankle rotation. This visualization highlights how the actuator’s motion results in controlled angular displacement at the output. To clarify the motion transmission mechanism, the “trajectory of motion between points A and B” refers to the path of the connecting link actuated by the stepper motor. In this context, point A represents the rotational output axis of the actuator (motor), while point B is a fixed pivot mounted on the prosthetic frame. This trajectory defines the input motion delivered to the joint mechanism and determines the angular displacement of the foot segment. The mechanism enables controlled motion in both the sagittal and frontal planes, as illustrated in Figure 4, Figure 5 and Figure 6.

Figure 5 shows the initial positions of points A and B with coordinates a_1_, b_1_ and a_2_, b_2_, as well as the geometric relationships between them. Foot rotation is calculated as the change in angle α between the axis of rotation and the line segment AB.

*(a)* 
*Coordinate Transformation and Point Rotation*


This section addresses the rotation of the shank segment relative to a fixed axis and the coordinate transformation that describes this motion. When the ankle joint rotates by an angle α, the position of each point on the foot changes. To mathematically model this movement, the initial trajectory coordinates (x, y) are transformed into a new coordinate system (x′, y′). This enables the calculation of the conjugate (rotated) trajectories of the points during angular motion. Figure 6 provides a visual representation of this transformation.

Here, *x*′ and *y*′ represent the coordinates of a virtual conjugate point, which mirrors the real trajectory path in a reference frame. This conjugate trajectory is introduced to evaluate the symmetry and dynamic stability of the foot’s motion. It helps assess how the system maintains balance during the inversion/eversion and dorsiflexion/plantarflexion phases.(1)$$\left\{\begin{array}{c} x^{\prime }=x\times cos\alpha +y\times sin\alpha \\ y^{\prime }=y\times cos\alpha -x\times sin\alpha \end{array}\right.$$
where

α—rotation angle of the shank joint;

x,y—coordinates of the experimental trajectory of the shank joint;

$x^{\prime },y^{\prime }$—coordinates of the conjugate trajectory.(2)$$x_{B}=x,\;\;y_{B}=-y_{i}$$(3)$$\left\{\begin{array}{c} {x^{\prime }}_{i}=\sqrt{x_{i}^{2}+y_{i}^{2}}\times \cos\left(2\alpha_{i}+arctg\left(\frac{y_{i}}{x_{i}}\right)\right)\\ y^{\prime }=\sqrt{x_{i}^{2}+y_{i}^{2}}\times \sin\left(2\alpha_{i}+arctg\left(\frac{y_{i}}{x_{i}}\right)\right) \end{array}\right.$$

In these equations, the index i represents the discrete steps of the simulation or individual configuration states within the ankle motion cycle. Its values range from i = 1  to n, where n is the total number of evaluated positions. This clarification helps in interpreting the sequential motion path and corresponding output parameters.

*(b)* 
*Geometric Configuration and Trajectory Parameters*


This section focuses on determining the spatial positions of points A, B, and C, which play a crucial role in the exoskeleton mechanism. The coordinates of these points are calculated based on their initial positions and the rotation angle (α). Using geometric relationships and trigonometric ratios, intermediate points along the motion trajectory are determined. This facilitates models of the full range of motion and the workspace of the system. Figure 6 and Figure 7 illustrate these geometric relationships.

Point T denotes the anatomical center of the talocrural joint, which serves as the main rotational axis for sagittal plane movements such as dorsiflexion and plantarflexion.

(4)$$\left\{\begin{array}{c} x_{c}=\alpha_{i}+d\\ y_{c}=y_{B}+\sqrt{h^{2}-({x_{B}-x_{A})}^{2}} \end{array}\right.$$(5)$$\left\{\begin{array}{c} x_{A}=\sqrt{\left(a_{1}^{2}+a_{2}^{2}\right)+\left(a_{1}^{2}+a_{2}^{2}\right)}\times \cos\left(2\alpha_{i}+arctg\frac{b_{1}+b_{2}}{a_{1}+a_{2}}\right)+x_{B}-{x^{\prime }}_{i}\\ y_{A}=\sqrt{\left(a_{1}^{2}+a_{2}^{2}\right)+\left(a_{1}^{2}+a_{2}^{2}\right)}\times \sin\left(2\alpha_{i}+arctg\frac{b_{1}+b_{2}}{a_{1}+a_{2}}\right)+x_{B}-{x^{\prime }}_{i} \end{array}\right.$$
where

$a_{1},b_{1\;}$—initial coordinates at point B;

$a_{2},b_{2}$—initial coordinates at point A.

*(c)* 
*Dynamic Equilibrium and Torque Analysis*


This section examines the dynamic model of the ankle exoskeleton. Specifically, it includes the calculation of the torque at point B (Md), the moment arm (dh), as well as the effects of external forces and accelerations. The modeling is based on Newton’s second law for rotational motion and considers gravitational forces, reaction forces, and the inertial properties of the system. These equations allow for an evaluation of the system’s stability, response underload, and efficiency in achieving the desired motion. Figure 8 illustrates the force and torque equilibrium diagram within the system.

Point E represents the effective contact location between the prosthetic foot and the ground during mid-stance. It is the point of application for the vertical ground reaction force in the finite element simulation.

Md is the torque of the ball joint at point B.(6)$$dh=\frac{\left|\left(y_{c}-y_{A}\right)\times x_{B}+\left(x_{A}-x_{c}\right)\times y_{B}+x_{c}\times y_{A}-y_{c}\times x_{A}\right|}{\sqrt{{\left(y_{c}-y_{A}\right)}^{2}+{\left(x_{A}-x_{c}\right)}^{2}}}$$$${\ddot{a}}_{i}=\varepsilon_{i}$$$$F_{h}\times dh+G\times \left(x_{ƶ}-x_{B}\right)\;\;\;\;\;\;\;\;\;\;\;\;\;\;\;\;\;\;\;\;\;\;\;\;=F_{EX}\times \left(y_{B}-y_{E}\right)+F_{Ey}\times \left(x_{E}-x_{B}\right)+y{\ddot{a}}_{i}\;\;\;\;\;\;\;\;\;\;\;\;\;\;\;\;\;\;\;\;\;\;\;\;-F_{EY}\times \left(XE-x_{B}\right)-F_{EX}\times \left(x_{E}-x_{B}\right)+y{\ddot{a}}_{i}$$$$M\alpha w\pm \bar{Fh}\times \bar{Uh}=0\;F_{h}\times d_{h}-F_{TX}\times \left(y_{B}-y_{T}\right)+F_{Ty}\times \left(x_{B}-x_{T}\right)+G\times \left(x_{ƶ}-x_{B}\right)=y{\ddot{a}}_{i}\;M\alpha w\pm \bar{Fh}\times \bar{Uh}=0$$
where

${\ddot{a}}_{i}-angular\;acceleration\;at\;point\;A$;

y−moment of rotational inertia at point B.

### 2.3. Design Using CAD Model

SolidWorks 2024 was used for the design of the active ankle prosthesis, enabling comprehensive three-dimensional modeling of all components and a preliminary load analysis of the key structural elements. The main design stages included the development of the overall device architecture, the selection of actuation and support components, and dimensional optimization to ensure biomechanical compatibility with the user.

Figure 9 shows the assembled CAD model of the prosthesis, including its main functional units: the foot, the actuation mechanism, and the mounting elements. The design allows for dorsiflexion and plantarflexion within a specified range of motion.

The labeled components in Figure 9 correspond to specific structural and functional parts of the 2-DoF ankle prosthesis. Label A refers to the upper support structure that serves as the main load-bearing element of the device. Point B indicates the rotational axis of the joint, allowing for dorsiflexion and plantarflexion during gait. Label C denotes the actuator block, which houses the motor responsible for driving joint motion. Point D marks the supporting linkage that contributes to mechanical stability and transmits force from the actuator. Label E represents the lead screw or linear actuator shaft, which converts rotary motion into linear displacement. Finally, point T corresponds to the toe end of the prosthetic foot, representing the primary ground contact area during walking. These labeled elements facilitate the understanding of the mechanical configuration and motion behavior of the prosthesis.

To ensure biomechanical compatibility and precise alignment with the anatomical parameters of the lower limb, the dimensions of the structure were optimized. Figure 10 presents the main overall dimensions of the model in three views: front, side, and top.

The overall dimensions of the prosthesis structure were optimized to ensure biomechanical compatibility with the parameters of the user’s lower limb and to replicate a natural gait pattern. During the modeling process, the following key dimensions were established (Figure 10):

The total height of the device is 420.7 mm, which corresponds to the anatomical proportions of the shank and allows for the proper integration of the actuation mechanism and mounting elements. The foot length from heel to toe is 295.2 mm, providing the necessary support during foot placement and step formation. The forefoot width is 80 mm, contributing to stability during the stance phase.

The platform height, measured from the bottom of the foot to the upper plate where the actuator module is located, is 296.1 mm, allowing for the compact placement of the actuation and sensor components. Additionally, the distance between the mounting axes in the upper part of the structure is 148.3 mm, ensuring a secure connection to the upper segments of the prosthesis and enabling future integration with the control system.

These parameters were selected based on an analysis of biomechanical requirements and the geometric constraints of the human lower limb, enabling a high ergonomic performance and structural reliability (see Table 3).

Figure 11 presents an exploded view of the design, showing all of the key components and standard parts used. The structure includes a NEMA 17 stepper motor [17HS15-1704S] (MybotOnline, Nanjing, China), an SFU1204-300 ball screw mechanism (JLD Bearing Co., Ltd., Lishui, China), fastening elements (mounts, bolts), as well as the base of the foot and the actuator module.

The key structural components include the following:–Foot base plate—the structural element of the foot platform that supports ground contact and connects to the lower actuator assembly.–Shank linkage—mechanical linkage that transmits motion from the actuator to the ankle joint.–Nema 17—stepper motor is responsible for generating the actuation motion.–SFU washer back/SFU1204-300—ball screw transmission that converts rotational motion into linear displacement.–Motor mount bracket, lead screw support bracket, and upper mount bracket—fastening components used to attach the drive mechanism (motor and screw) to the foot structure and the shank linkage.–Hex flange bolt—standard fasteners used to secure and stabilize movable mechanical parts.

The developed design provides the required range of motion consistent with biomechanical demands, while also enabling the integration of a closed-loop control system, including position and force sensors. The modular layout allows for the easy modification or replacement of individual components, facilitating customization to the specific needs of the user.

### 2.4. Mechanical Structure

To develop an autonomous active ankle prosthesis with two degrees of freedom, an experimental mechanical structure was designed, integrating movable joint elements, the actuation system, and the control module. The prototype was assembled under laboratory conditions for subsequent functional testing and control algorithm debugging.

Figure 12 shows the overall view of the physical model of the device. The structure includes the lower foot platform, ankle joint assembly, shank segment, stepper motor, ball screw transmission, as well as the housing and control module based on the ESP32-CAM.

To provide a clear understanding of the structural layout, Figure 13 presents a schematic diagram highlighting the key components.

The design was developed using a modular approach, allowing for the easy replacement or adaptation of components depending on the required degrees of freedom. The inversion/eversion and dorsiflexion/plantarflexion mechanisms are implemented through a stepper motor actuator connected to a ball screw drive, which converts rotational motion into linear displacement. This linear motion is then transformed into the angular movement of the foot relative to the shank.

In study [22], the NEMA 17 stepper motor was used for the first time in the design of a rehabilitation ankle exoskeleton. The choice of this motor was based on its reliability, compact size, and sufficient torque to drive the prototype. The device is controlled via a Bluetooth connection, allowing the rehabilitation specialist to remotely operate the exoskeleton during therapy sessions.

The device is controlled by an ESP32-CAM microcontroller, which not only manages motor operation but also enables wireless data transmission, video monitoring, and integration with various sensors (IMU, encoders, etc.). This configuration makes the prosthesis suitable for laboratory research with the potential for further development and functional expansion.

The mechanical structure is specifically designed to provide the following:–The required range of motion within physiological limits (±20–25 deg.).–Structural strength and frame rigidity underload.–Compact dimensions for ergonomic use.–Ease of assembly and modification.

Thus, the presented prototype of the active prosthesis serves as a functional platform for further testing and demonstrates the feasibility of achieving two degrees of freedom using low-cost and readily available components.

Preliminary estimates of power consumption were calculated based on actuator torque and velocity profiles during a typical gait cycle. The average electrical power required per step was found to be approximately 12–15 W. Considering a battery pack composed of three 18,650 lithium ion cells (each rated at 3.7 V and 3000 mAh), the total energy capacity is approximately 33.3 Wh. This configuration allows for an estimated continuous operation time of 2.2–2.7 h under standard walking conditions. These values support the claim of low power consumption and demonstrate the feasibility of high autonomy in practical scenarios.

## 3. Results

This section presents the results of the numerical simulations, including the kinematic analysis of inversion/eversion and dorsiflexion/plantarflexion movements, as well as static strength analysis using the Finite Element Method (FEA).

### 3.1. Simulation—Inversion/Eversion Movement

To evaluate the effectiveness of the developed autonomous active ankle prosthesis with two degrees of freedom in the frontal plane, a simulation of inversion and eversion movements was performed using the complete CAD model shown in Figure 1. The motion of the foot support plate relative to the shank segment was achieved through an SFU1204-300 screw drive, actuated by a NEMA 17 stepper motor, providing precise lateral tilting movement.

Figure 14 illustrates the behavior of the ankle joint in three distinct positions: full inversion, neutral, and full eversion. These positions demonstrate the prosthesis’s ability to replicate natural movements in the frontal plane, which are critical for maintaining balance especially when walking on uneven terrain. Inversion and eversion control was achieved by adjusting the angular displacement of the screw, generating the required torque through the connection between the shank and foot.

The angular displacement range during the simulation was set within ±10°, corresponding to physiological norms. The CAD-based simulation results confirmed that the design enables smooth rotation around the frontal axis without mechanical jamming or misalignment in the joints.

Figure 15 presents the input and output signals of the system during the simulation of ankle joint inversion/eversion movement.

The input signal was defined as a sinusoidal angular displacement trajectory with an amplitude of ±10° and a period of approximately 6 s. This function represents the desired behavior of the prosthesis during foot movement in the frontal plane.

The output signal, measured from the end effector (foot support plate), closely follows the shape of the input signal, indicating adequate dynamic performance of the system and high tracking accuracy. The maximum deviation reached approximately ±11°, which reflects minor force redistribution within the mechanism due to inertial and elastic factors, without compromising actuator functionality.

The output waveform remains stable and symmetric with respect to the time axis, confirming the balanced operation of the structure and the effective transmission of rotational torque from the stepper motor through the screw drive to the foot segment.

These results demonstrate the system’s ability to accurately reproduce the intended inversion and eversion kinematics, which is critically important for maintaining stability and adaptability while walking over uneven terrain.

An additional analysis was conducted on the trajectory of the system’s center of mass (CoM) during the inversion/eversion motion. Figure 16 shows the CoM position over time along the X- and Y-axes.

As seen in the graph, periodic oscillations along the X-axis occur with an amplitude of approximately ±50 mm, corresponding to the lateral tilting of the foot during inversion and eversion. This horizontal movement pattern is expected and demonstrates the natural mass shift in the prosthesis during frontal plane motion.

Along the Y-axis (vertical component), deviations are minimal within ±5 mm from the average position of around 270 mm. This indicates a high level of vertical stability during motion execution, which is essential for maintaining overall balance and preventing vertical instability.

Thus, the simulation confirms that the active actuator can accurately control the position of the center of mass in the frontal plane without significant vertical fluctuations. This demonstrates the reliability of the design and its suitability for real-world applications such as walking on inclined or uneven terrain.

An additional evaluation of the system’s linear displacement in the frontal plane was conducted to assess kinematic stability during inversion/eversion movement. The results are presented in Figure 17, which shows the displacement trajectories along the X- and Y-axes over time.

As seen in the graph, the X-axis displacement exhibits harmonic oscillations with an amplitude of approximately ±50 mm, corresponding to the expected lateral movement of the foot during inversion and eversion. This confirms that the system accurately transmits the translational motion generated by ankle joint rotation.

In the Y-axis direction (vertical), the displacement remains at least less than ±5 mm from the average position of about 250 mm. This indicates a high level of vertical stability and effective compensation for inertial loads during movement.

Thus, the linear displacement data confirms the stable operation of the actuation mechanism under frontal plane dynamics and demonstrates that the resulting motion meets the biomechanical requirements for an ankle prosthesis.

To further analyze the kinematic behavior of the prosthesis, the velocity of the foot support plate along the X-axis over time was examined. Figure 18 presents the graph of linear velocity in the frontal plane corresponding to the inversion/eversion movement.

The graph exhibits the characteristics of harmonic oscillatory motion, with a maximum velocity amplitude of approximately ±9 mm/s. Peak values occur around the 1st and 5th seconds, corresponding to the phases of rapid foot tilt. Between these peaks, zero-velocity intervals are observed, marking the moments when the direction of motion changes.

This velocity curve confirms that the system reproduces smooth and controlled lateral movement without abrupt jerks or unstable accelerations. The absence of high-frequency fluctuations indicates reliable actuator performance and sufficient accuracy in real-time position regulation.

Thus, the obtained data confirms the mechanism’s ability to accurately implement the required inversion/eversion kinematics with controlled speed, an essential factor for ensuring the safety and adaptability of the prosthesis under dynamic conditions.

To complete the analysis of inversion/eversion dynamics, the graph of vertical velocity (Y-axis) was examined, as shown in Figure 19.

As evident from the graph, the vertical motion follows a harmonic pattern but with an extremely small amplitude of approximately ±0.045 mm/s. This indicates a high level of structural stability in the vertical plane and the absence of significant vibrations or oscillations that could affect user comfort or safety.

Velocity peaks occur around the 3rd and 6th seconds; however, their magnitudes remain well within acceptable biomechanical limits. This minimal vertical displacement confirms that inversion and eversion movements do not disrupt the system’s balance in the vertical direction.

The data reflects a well-balanced design and the effective kinematic decomposition of motion: active movement is confined to the intended plane (frontal) without unwanted responses in other degrees of freedom.

To conclude the analysis, the angular kinematics of the active ankle joint were examined using Euler angles. Figure 20 shows the system’s rotation angle over time, describing the motion of the foot segment relative to the shank segment during inversion and eversion.

The graph displays a typical sinusoidal shape, corresponding to harmonic angular displacement with an amplitude of approximately ±11 deg. Peak values occur around 1.8 and 4.8 s, reflecting the extreme inward (inversion) and outward (eversion) tilt positions of the foot.

This curve shape indicates the accurate execution of the intended motion and effective system tracking of the reference trajectory. The absence of abrupt spikes or nonlinear phases confirms kinematic continuity and coordination between all mechanical components (motor, screw, and linkage elements).

Thus, the analysis of Euler angles confirms that the developed system provides smooth, controlled, and biomechanically accurate movement, essential for functional ankle joint prosthetics.

To conclude the analysis, a generalized curve of foot tilt angle variation was constructed, as shown in Figure 21. This graph illustrates the resulting angular position of the end effector relative to the shank segment throughout the entire inversion and eversion movement cycle.

The curve follows a sinusoidal pattern with an amplitude of approximately ±11 deg., confirming alignment with the physiological range of motion in the frontal plane. Over the full cycle lasting about 6 s two primary peaks are observed, corresponding to the maximum inversion and eversion positions.

The smooth shape of the signal indicates the absence of jerks or abrupt transitions, confirming the coordinated operation of the entire mechanism from the motor and screw drive to the final kinematic chain. Moreover, the lack of distortions or phase shifts between the reference and actual angles demonstrates the system’s high precision and stability.

In summary, the combined analysis of all presented graphs confirms that the developed active ankle joint design is capable of accurately, reliably, and biomechanically performing inversion and eversion movements under real-world operating conditions.

The simulation results confirm that the proposed 2-DoF autonomous active ankle–foot prosthesis performs the inversion/eversion movement accurately and stably. The angular displacement reaches ±11 deg., which corresponds well with physiological limits. Linear and angular velocities are smooth and periodic, with no indication of abrupt transitions or instability.

Notably, the center of mass (COM) maintains vertical stability (±5 mm), while lateral displacement occurs as expected. Vertical velocity is minimal, confirming the structural robustness of the system.

Overall, the model demonstrates that the prosthesis is capable of mimicking natural frontal plane ankle motion, making it suitable for use in dynamic environments requiring balance on uneven terrain.

### 3.2. Simulation—Dorsiflexion and Plantarflexion Movement

To evaluate the performance of the developed two-degree-of-freedom active ankle prosthesis in the sagittal plane, a simulation of dorsiflexion (upward flexion) and plantarflexion (downward flexion) was conducted. These movements play a crucial role during the toe-off and landing phases of walking, stair climbing, and other daily activities.

Figure 22 presents the CAD model of the prosthesis in three different positions: maximum dorsiflexion (left), neutral (center), and maximum plantarflexion (right). Motion control was achieved through the SFU1204 ball screw mechanism driven by a NEMA 17 stepper motor. The kinematic structure ensured highly accurate torque transmission without jamming or mechanical interference.

The simulated range of angular tilt was approximately ±11 deg., which aligns with physiological values for sagittal plane motion during level ground walking. The mechanism demonstrated smooth and stable operation, and the structure proved to be both reliable and biomechanically appropriate.

Thus, the simulation confirms that the developed system can accurately reproduce natural foot movements in the longitudinal plane, an essential requirement for energy-efficient propulsion and impact absorption during locomotion.

An additional analysis was conducted on the linear displacement dynamics of the system along the Y-axis (vertical direction) during dorsiflexion and plantarflexion. Figure 23 presents the vertical displacement of the foot over time during one complete movement cycle.

The graph demonstrates harmonic motion within a range from –120 mm to –90 mm, corresponding to the vertical raising and lowering of the foot resulting from sagittal plane rotation. The minimum value occurs during maximum plantarflexion (toe pointing downward), while the maximum is reached during dorsiflexion (toe lifting upward).

This vertical displacement is essential for replicating the natural heel-off phase during walking, as well as for impact absorption during landing. The absence of sharp spikes and the smooth curvature of the graph indicate the reliable performance of the kinematic mechanism and precise force transmission.

To assess structural stability during motion in the sagittal plane, the trajectory of the system’s center of mass (CoM) was further analyzed. Figure 24 shows the CoM position over time along the Y- (vertical) and Z- (longitudinal) axes.

The graph indicates that along the Y-axis, the center of mass remains nearly constant within a range of approximately ±10 mm, oscillating around an average of 280 mm. This suggests a high vertical stability of the structure during upward and downward foot movements (dorsiflexion/plantarflexion).

More pronounced changes are observed along the Z-axis, ranging from 0 to –160 mm, reflecting the forward and backward motion of the entire foot along the longitudinal axis. This displacement corresponds to the swing and push-off phases of gait and is an expected outcome of dorsiflexion and plantarflexion.

The CoM dynamics confirm that the system maintains stability during sagittal plane operation and that the motion mechanics provide balanced kinematics without a loss of equilibrium.

An additional parameter in the analysis of dorsiflexion and plantarflexion movement is the angular velocity of foot rotation. Figure 25 presents the time-dependent angular velocity in degrees per second (deg/s) over the course of one complete movement cycle.

The graph shows a harmonic rotation pattern, with alternating positive and negative values corresponding to the phases of foot lifting and lowering. The peak angular velocity reaches approximately ±2.1 deg/s, which falls within the range of typical physiological speeds for the ankle joint during normal walking.

A notable feature of this graph is the smooth transition between phases and the absence of abrupt spikes, indicating stable actuator performance and good trajectory control. Additionally, the oscillations are synchronized with the primary gait frequency, further confirming the realism of the simulated motion.

Overall, the presented data demonstrate that the system effectively controls angular velocity during foot flexion and extension, delivering biomechanically sound movement and enhanced user comfort.

To complete the analysis of sagittal plane motion, the linear velocity of the system along the X- and Z-axes was examined, as shown in Figure 26.

Along the X-axis (lateral horizontal direction), harmonic oscillatory motion is observed with an amplitude of up to ±1.5 mm/s. This corresponds to indirect lateral displacements caused by the plantarflexion and dorsiflexion of the foot. The velocity peaks align with the peak moments of angular motion, indicating consistency among the kinematic parameters.

Along the Z-axis (longitudinal axis forward/backward), less pronounced oscillations are seen, reaching up to ±0.5 mm/s, reflecting the reciprocating motion during step simulation. These values are particularly important for analyzing motion during the foot’s swing and landing phases.

Both axes exhibit stable and smooth dynamic behavior without abrupt spikes, confirming the proper implementation of plantarflexion/dorsiflexion through the mechanical system and control algorithm.

To evaluate the angular kinematics during sagittal plane movement, Euler angles were calculated and presented in Figure 27. The graph illustrates the change in foot orientation relative to the shank during dorsiflexion and plantarflexion.

The curve shows a periodic variation in angle from 0 deg. to approximately 21 deg., with two complete cycles over a 6 s interval. Peak values occur during the phases of maximum flexion and extension, corresponding to the extreme foot positions during walking.

A key feature is the smooth trajectory, the absence of discrete jumps, and the symmetry of the signal, all of which confirm the precise tracking of the desired trajectory and balanced mechanical motion.

Thus, the analysis of Euler angles demonstrates that the system accurately transmits rotational movements in the sagittal plane, closely aligning with anatomically correct values for dorsiflexion and plantarflexion.

The final component of the dorsiflexion and plantarflexion analysis is the overall graph of foot angular deviation over time, shown in Figure 28.

The graph demonstrates harmonic variation in the angle within a range from −22 deg. to +20 deg., covering the full physiological range of motion in the sagittal plane. This angular displacement enables the execution of both the extension (plantarflexion) and flexion (dorsiflexion) phases essential for normal gait.

The smooth curve profile, free of abrupt spikes, confirms the stable operation of the mechanism, the effectiveness of stepper motor control, and the kinematic compatibility of all system components. The temporal symmetry of the graph further indicates proper load transmission and accurate motion control.

Thus, the presented results confirm that the design is capable of accurately reproducing ankle joint biomechanics in dynamic conditions.

Below is a summary table presenting the results of the numerical simulation of foot motion in the sagittal plane, including the displacement parameters, velocity, center of mass position, and angular kinematics.

The obtained results confirm that the proposed design of the two-degree-of-freedom active ankle prosthesis provides the biomechanically accurate execution of dorsiflexion and plantarflexion movements. The angular displacement amplitude reaches ±20–22°, which corresponds to physiological values during level ground walking.

The center of mass maintains vertical stability, while longitudinal displacement exhibits the expected dynamics during the swing and push-off phases. Both angular and linear velocities remain within safe and controllable limits, ensuring smooth motion without abrupt transitions.

Thus, the system demonstrates a high accuracy, stability, and efficiency, making it a promising solution for use in lower limb active prostheses with adaptability to real-world locomotion scenarios.

### 3.3. FEA

To evaluate the structural performance and reliability of the design, a finite element analysis (FEA) was carried out using SolidWorks 2024 Simulation. The analysis was conducted under static loading conditions to determine the stress distribution, deformation behavior, and strain levels under forces typical for the stance phase of gait.

The model was constrained at the shank attachment interface to replicate the fixation of the prosthesis to the residual limb. A vertical load simulating the ground reaction force (GRF) during mid-stance was applied to the foot sole, representing the phase when the entire body weight is supported by the stance leg.

A tetrahedral meshing was applied to achieve an accurate geometry representation and proper stress evaluation in critical areas (Figure 29). A finer mesh was used near the joints, screw mechanism, and foot attachment areas to improve calculation accuracy.

The load-bearing components—NEMA 17 stepper motor [17HS15-1704S] (MybotOnline, China), SFU1204 lead screw (JLD Bearing Co., Ltd., Lishui, Zhejiang Province, China), and Rod End M12 joint (from a supplier in China)—were made of AISI 304 stainless steel. AISI 304 is an austenitic stainless steel containing approximately 18% chromium and 8% nickel. It is widely used due to its excellent corrosion resistance, good formability, and sufficient strength for moderate load-bearing applications. These properties make it suitable for mechanical components exposed to both static loads and environmental factors. Its properties are as follows:–Yield strength—2.07 × 10^8^ N/m^2^.–Tensile strength—5.17 × 10^8^ N/m^2^.–Young’s modulus—1.9 × 10^11^ N/m^2^.–Shear modulus—7.5 × 10^10^ N/m^2^.–Poisson’s ratio—0.29.–Density—8000 kg/m^3^.–Thermal expansion coefficient—1.8 × 10^−5^ 1/K.

All other housing and fastening elements were modeled using ABS plastic (from a supplier in China), which reduces the overall weight while providing sufficient strength for non-critical parts.

The generated tetrahedral mesh generates a detailed geometry representation and accurate stress evaluation. A finer mesh was applied in joint regions and connection points to improve solution accuracy.

Figure 30 presents the von Mises stress distribution. The maximum equivalent stress was $\sigma_{max}=1.498\times 10^{8}$, which is below the yield strength of AISI 304, indicating that the structure remains in the elastic region. Stress concentrations were observed mainly near joint interfaces and abrupt cross-section transitions.

Figure 31 shows the total displacement of the structure under load, with a maximum value of approximately 39.5 mm, mainly occurring in the upper flexible elements.

Figure 32 illustrates the equivalent strain (ESTRN) distribution. The maximum strain reached approximately 0.013, confirming elastic behavior without material failure.


*FEA Results Summary*


The results of the FEA are as follows:

–Maximum von Mises stress: $1.498\;\times \;10^{8}\;N/m^{2}$.–Maximum total displacement: ≈39.5 mm.–Maximum equivalent strain: ≈0.013 (dimensionless).–Safety factor:$$SF\;=\frac{\sigma_{yield}}{\sigma_{max}}\;=\frac{2.07\;\times 10^{8}}{1.498\times 10^{8}}\;\approx 1.38$$

The results confirm that the structure withstands critical loads within the elastic range of AISI 304. However, the current analysis is limited to static conditions.

Future work will include the following:–The dynamic simulation of the full gait cycle (from heel strike to toe-off).–Fatigue strength assessment under cyclic loading.–A consideration of screw mechanism compliance and backlash in the actuator transmission model to improve simulation accuracy.

Flexible components, such as the spring element, were simplified as rigid parts made of ABS plastic in the static analysis. Their elastic compliance was not included in the current FEA and will be incorporated in the future dynamic simulation for more accurate results.

## 4. Conclusions

In this work, a concept of an autonomous active ankle prosthesis with two degrees of freedom (2-DoF) was developed, implemented, and analyzed. The prosthesis is capable of reproducing movements in both the sagittal plane (dorsiflexion and plantarflexion) and the frontal plane (inversion and eversion). The proposed device is aimed at improving stability, adapting to uneven terrain, and enhancing gait symmetry in users with transtibial amputation.

An optimized mechanical structure was designed using CAD modeling, incorporating a ball screw transmission driven by a NEMA 17 stepper motor. The system was controlled via an ESP32-CAM microcontroller, enabling an autonomous and modular architecture. Numerical analyses including kinematic motion simulation and finite element analysis (FEA) confirmed that the prosthesis provides physiologically appropriate angular ranges (±20–22 deg.) with high structural stability and reliability.

The simulation results demonstrated the effective execution of both frontal and sagittal movements without mechanical jamming, with minimal vertical displacement of the center of mass and smooth kinematics essential for maintaining user dynamic stability. The FEA results showed that the observed stress and strain levels remained within acceptable limits, not exceeding the elastic properties of the selected material (AISI 304 stainless steel).

It is important to note that the reported angular displacement range of ±20–22 deg., corresponds to the theoretical mechanical capabilities of the prosthesis in both motion planes, as verified through kinematic simulations. Meanwhile, the dorsiflexion angle of approximately 7 deg., observed in the FEA study, represents elastic deformation under static vertical loading and does not indicate the full actuation capacity of the system. This clarification ensures consistency between the mechanical design intent and the numerical modeling outcomes.

At this stage, mechanical testing of the materials under repeated loading and ground reaction forces (GRFs) has not yet been conducted. These evaluations are planned as part of future work once the final prototype is assembled. This will allow the verification of the long-term durability and real-world reliability of the prosthesis under gait-specific cyclic loading conditions.

Thus, the proposed system shows strong potential for practical use in lower limb active prosthetics. It is acknowledged that the current prototype was developed based on average anthropometric parameters and may not be directly compatible with all transtibial or transfemoral amputees, especially in cases involving residual limb asymmetry or longer-than-average limb stumps. Future development will include modularity and scalable design features to ensure a better fit and customization for a wider range of users, thereby enhancing clinical applicability.

Additionally, clinical and experimental gait analysis using force plates and motion capture tools has not yet been performed. These trials are planned in the next phase to validate the simulation outcomes and assess the real-world performance of the proposed prosthesis. Future work will focus on refining adaptive control algorithms based on sensory feedback, conducting laboratory and clinical testing, and exploring the integration of neural interfaces to enable intuitive control.

It should also be noted that the finite element simulations presented in this study assumed ideal axial loading, whereas in real prosthetic use, a residual limb socket interface introduces angular misalignments and non-axial forces. These factors may contribute to estimation errors that will be addressed in future mechanical testing. Additionally, the developed system integrates inertial measurement units (IMUs), force sensors, and encoders to ensure proper orientation tracking and real-time feedback for adaptive control. Further optimization and calibration of these sensors will be essential for improving prosthesis responsiveness and user safety.

Furthermore, clinical feasibility studies and real-world testing have not yet been conducted. These will be addressed in future work through functional trials with end-users to evaluate safety, usability, and performance under daily life conditions.

Additionally, it should be noted that the simulations conducted in this study are based on idealized assumptions, such as perfect sensor feedback and precise motor dynamics. In practice, various uncertainties may influence the prosthesis performance, including sensor noise, mechanical backlash in the screw transmission, and simplified modeling of the foot–ground interaction. These factors may cause deviations between the simulated and real-world results. Future work will incorporate these aspects by applying sensor fusion, refining the control algorithms, and validating performance through experimental gait trials under dynamic loading conditions.

This development may serve as a foundation for the creation of more accessible, autonomous, and biomechanically effective prosthetic solutions, bringing artificial limb functionality closer to the capabilities of a healthy ankle joint.

## Figures and Tables

**Figure 1 sensors-25-04881-f001:**
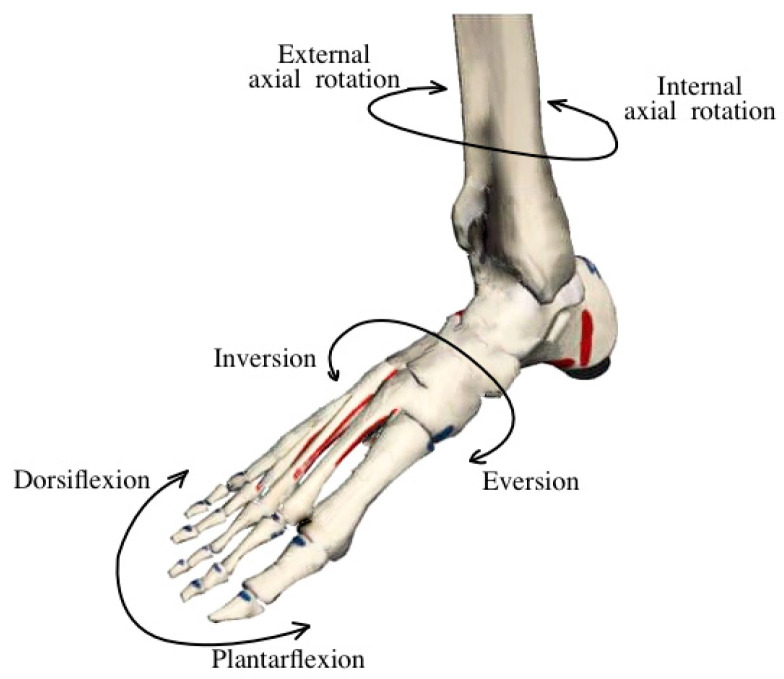
The main movements of the ankle joint in the sagittal and frontal planes [20].

**Figure 2 sensors-25-04881-f002:**
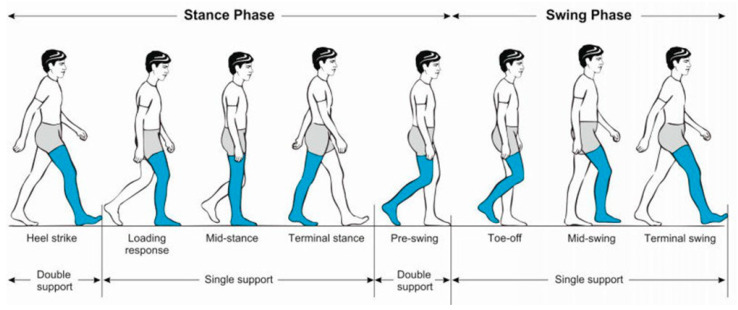
Main phases of the gait cycle and the role of the ankle joint [21].

**Figure 3 sensors-25-04881-f003:**
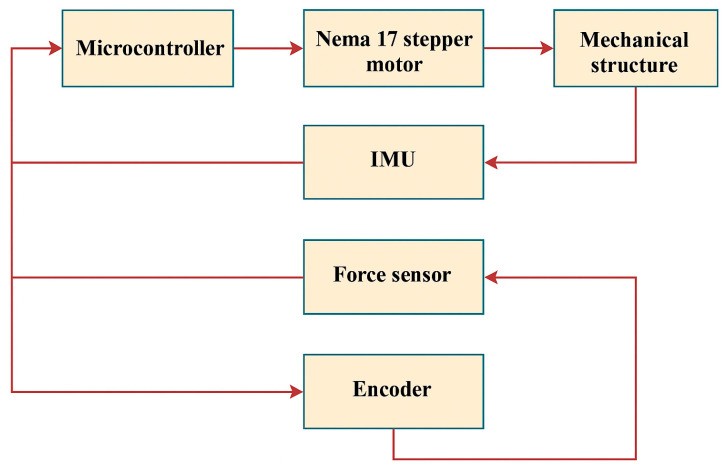
Architecture of the adaptive control system for an active prosthesis using sensors and controllers.

**Figure 4 sensors-25-04881-f004:**
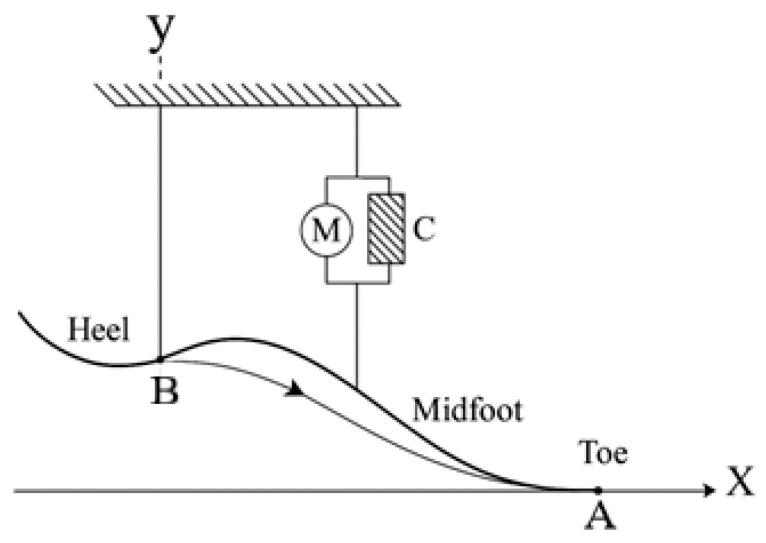
Initial coordinates and trajectories of points A and B.

**Figure 5 sensors-25-04881-f005:**
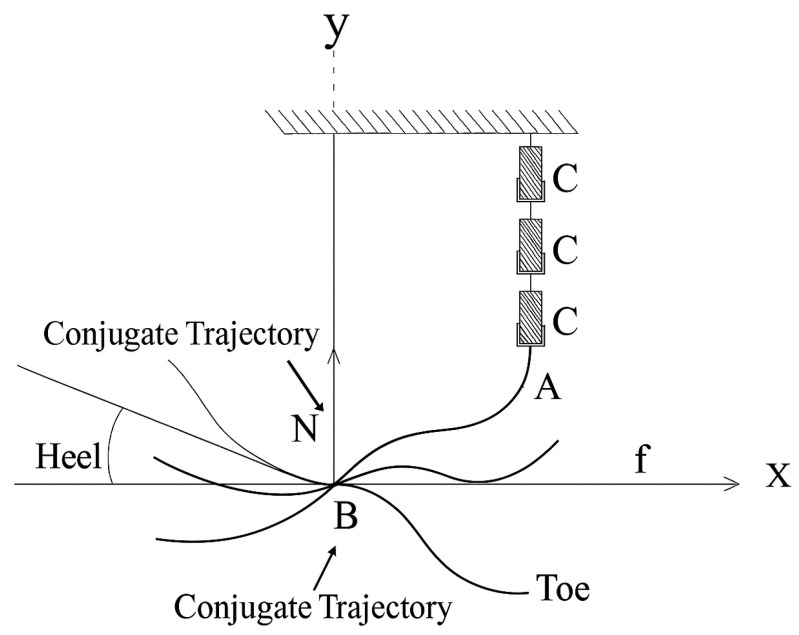
Ankle joint rotation angle α. A, B—key points of the foot; C—actuator/motor positions; N—normal vector; f—force direction; x, y—coordinate axes.

**Figure 6 sensors-25-04881-f006:**
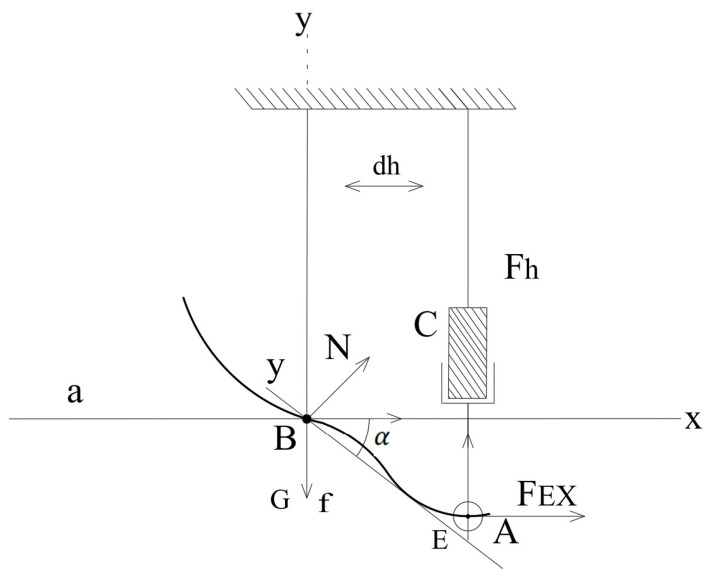
Conjugate trajectory and angular motion with coordinates $x^{\prime }i,\;y^{\prime }i$.

**Figure 7 sensors-25-04881-f007:**
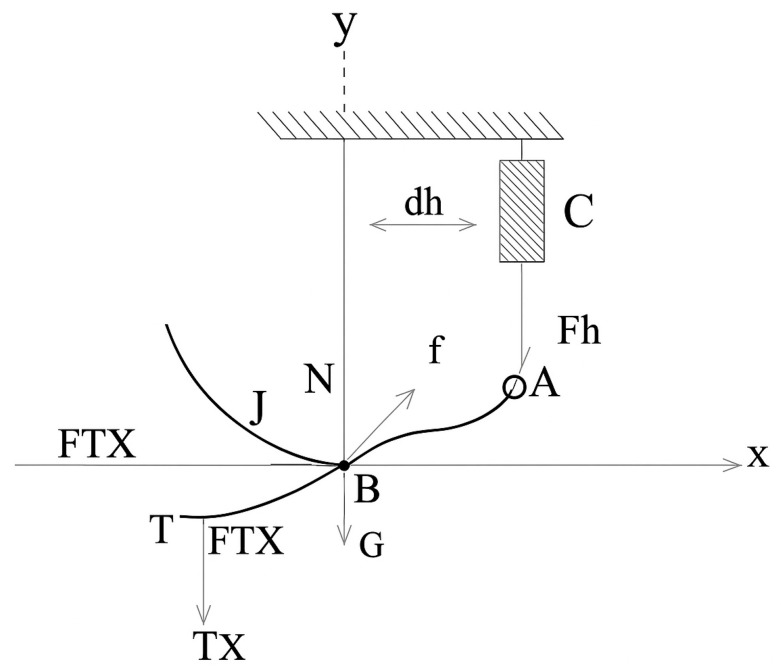
Relationship between points A and C and calculation of coordinates y_c, x_c.

**Figure 8 sensors-25-04881-f008:**
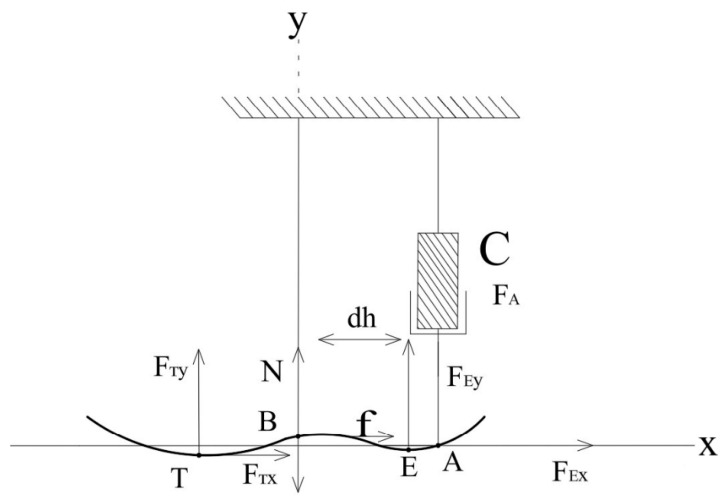
Load and torque diagram (dynamic model).

**Figure 9 sensors-25-04881-f009:**
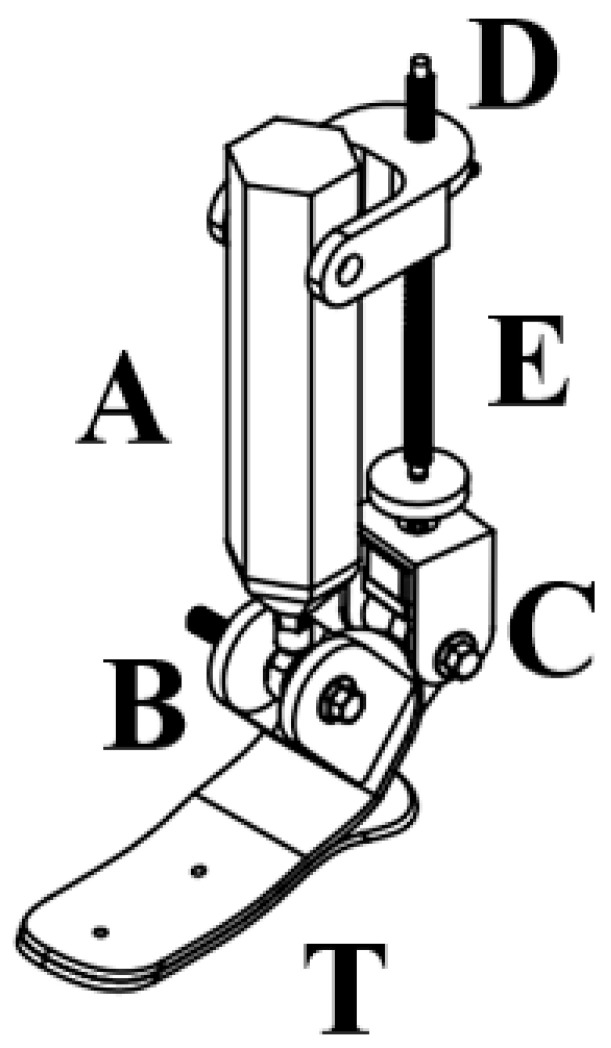
Annotated CAD model of the 2-DoF active ankle prosthesis. The key points are labeled as follows: A—upper support structure, B—ankle joint axis, C—motor unit, D—supporting link, E—screw actuator rod, T—toe section of the foot.

**Figure 10 sensors-25-04881-f010:**
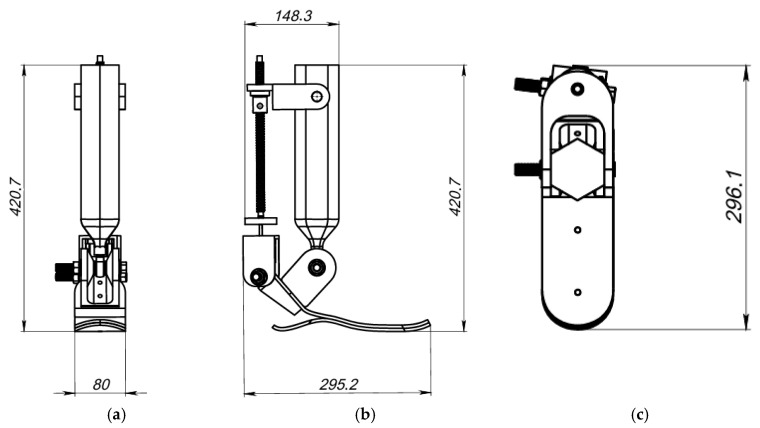
Overall dimensions of the prosthesis in different views: (**a**) front view; (**b**) side view; (**c**) top view (dimensions in mm).

**Figure 11 sensors-25-04881-f011:**
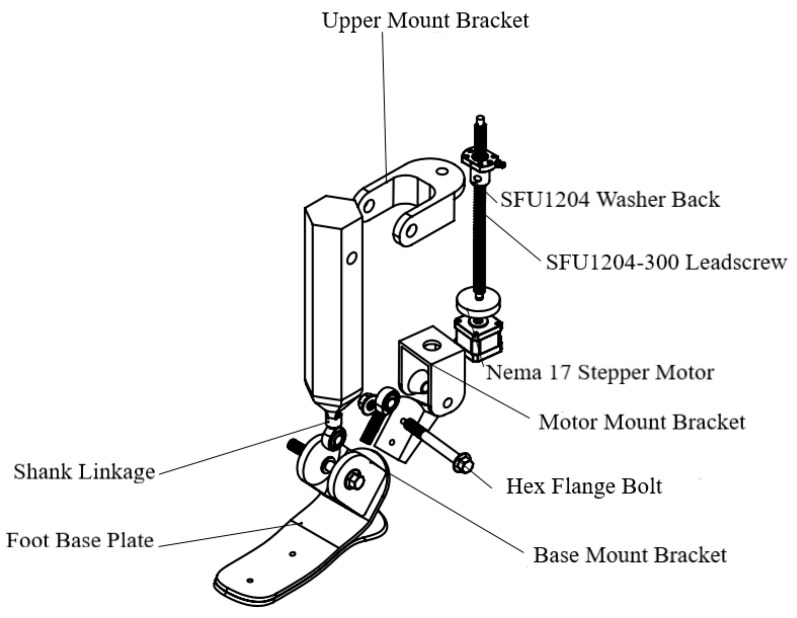
Exploded CAD model of the ankle exoskeleton with key mechanical components.

**Figure 12 sensors-25-04881-f012:**
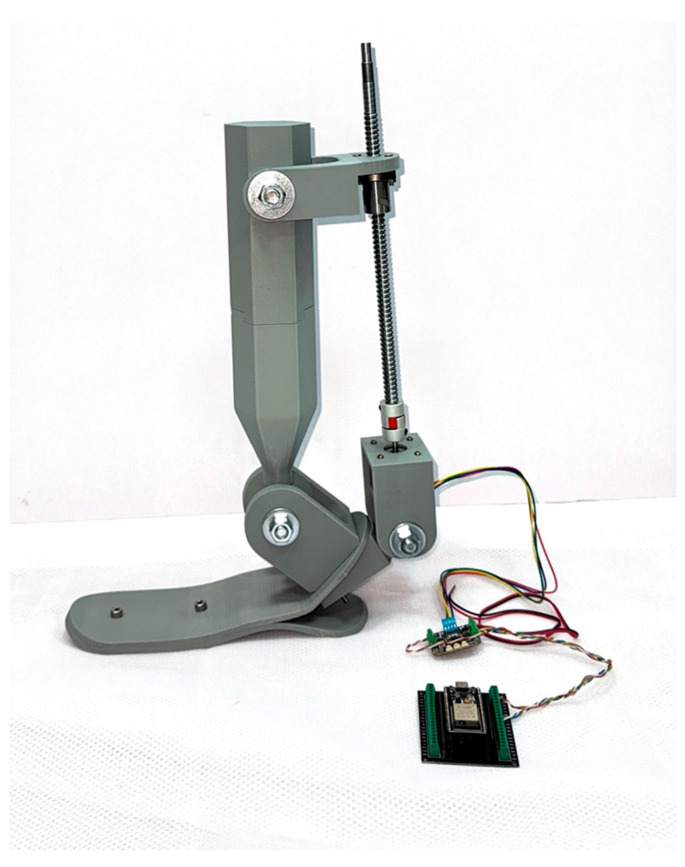
General view of the experimental prosthesis with actuation mechanism.

**Figure 13 sensors-25-04881-f013:**
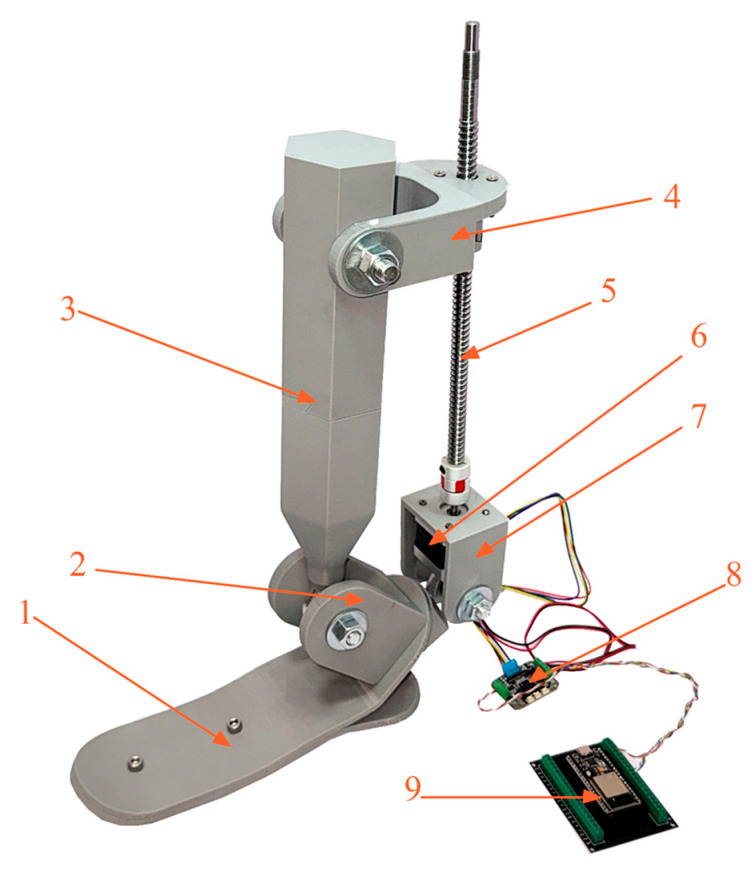
Main structural components of the prosthesis: 1—foot platform; 2—hinge joint (axis of rotation); 3—shank segment; 4—mounting plate; 5—ball screw drive (SFU1204-300); 6—coupling connector; 7—motor housing; 8—stepper motor (NEMA 17); 9—control unit (ESP32-CAM).

**Figure 14 sensors-25-04881-f014:**
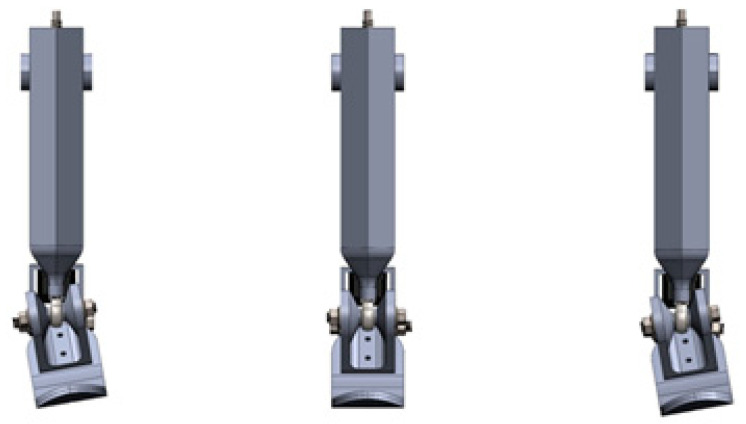
The CAD model shows the ankle–foot prosthesis in three positions: full inversion (**right**), neutral (**center**), and full eversion (**left**).

**Figure 15 sensors-25-04881-f015:**
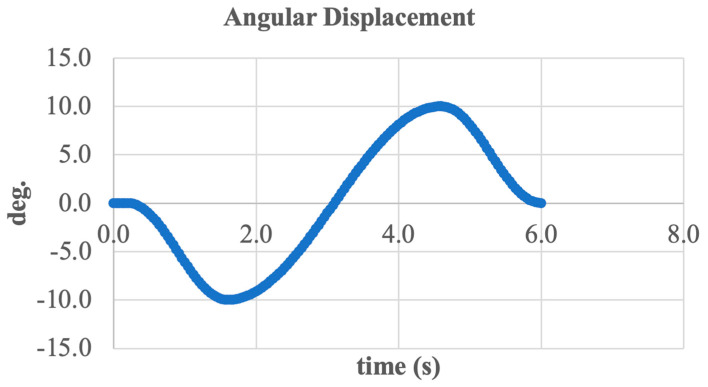
This is a figure. Schemes follow the same formatting.

**Figure 16 sensors-25-04881-f016:**
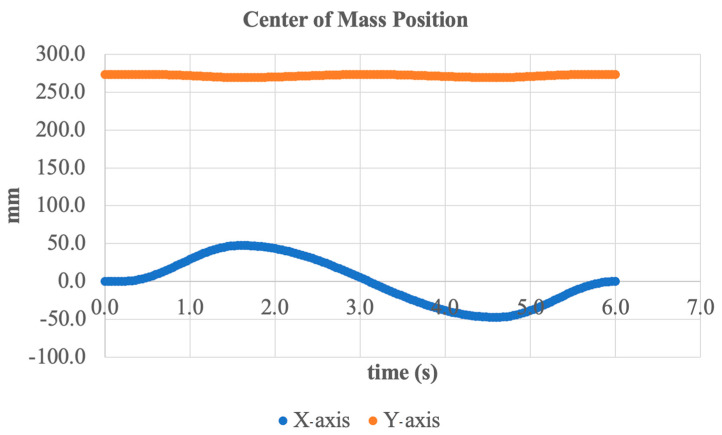
Center of mass position of the system along X- and Y-axes during inversion/eversion.

**Figure 17 sensors-25-04881-f017:**
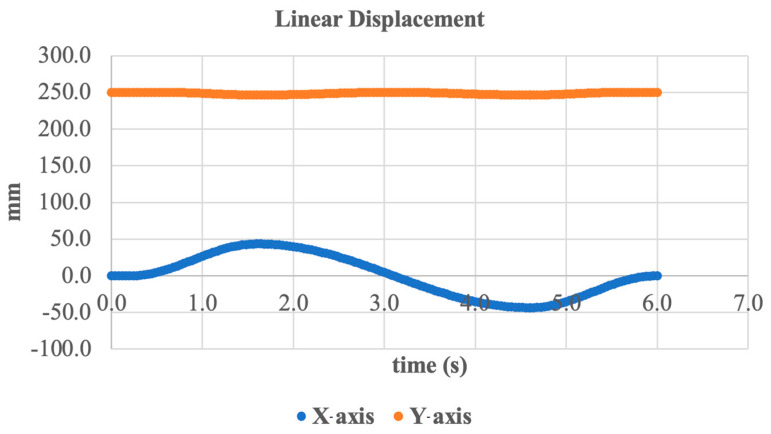
Linear displacement of the foot segment along X- and Y-axes.

**Figure 18 sensors-25-04881-f018:**
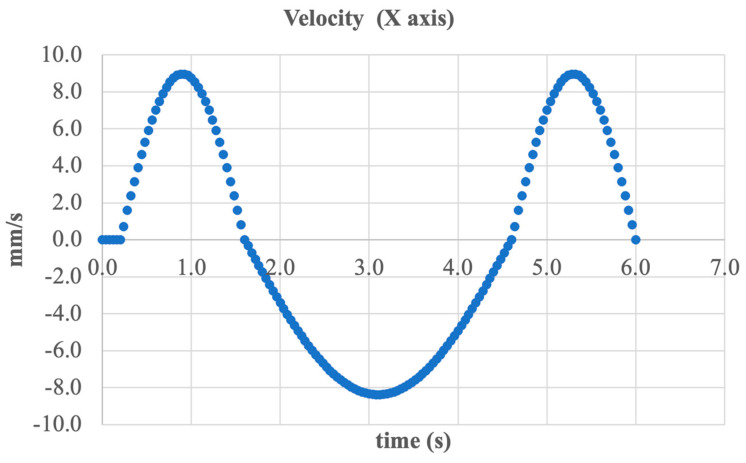
Linear velocity of the foot segment along the X-axis.

**Figure 19 sensors-25-04881-f019:**
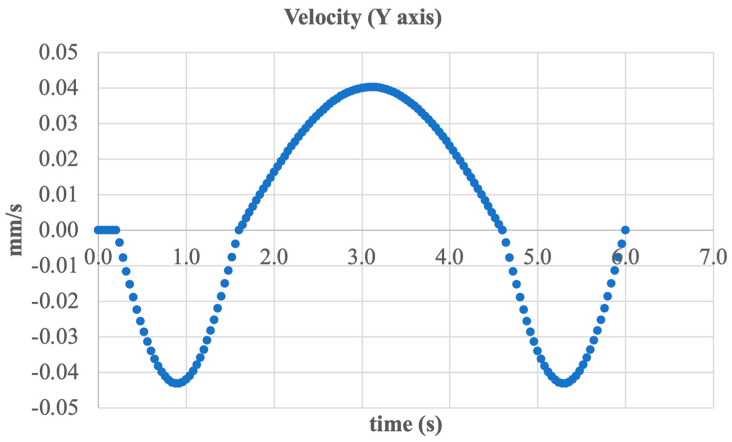
Linear velocity of the foot segment along the Y-axis.

**Figure 20 sensors-25-04881-f020:**
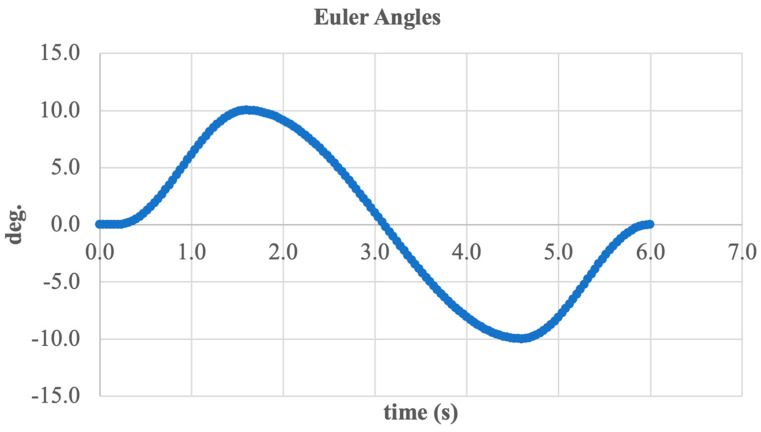
Euler angle variation in the ankle joint during inversion/eversion.

**Figure 21 sensors-25-04881-f021:**
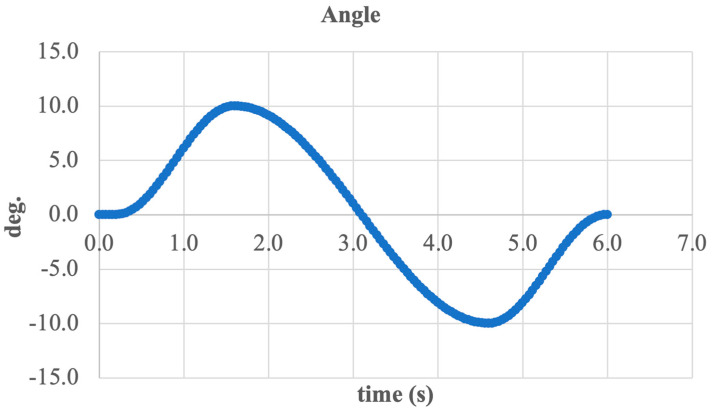
Overall angular position of the foot during inversion/eversion cycle.

**Figure 22 sensors-25-04881-f022:**
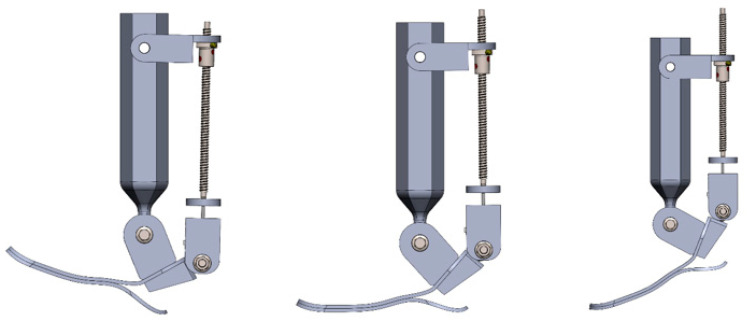
CAD model of the ankle–foot prosthesis in dorsiflexion (**left**), neutral (**center**), and plantarflexion (**right**) positions.

**Figure 23 sensors-25-04881-f023:**
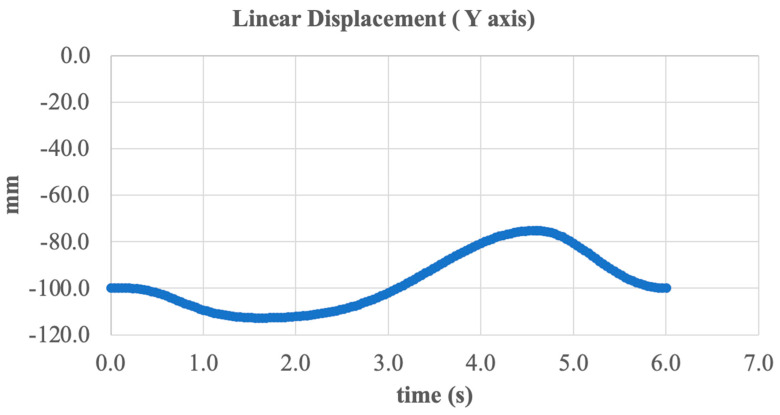
Vertical linear displacement of the foot during dorsiflexion/plantarflexion (Y axis).

**Figure 24 sensors-25-04881-f024:**
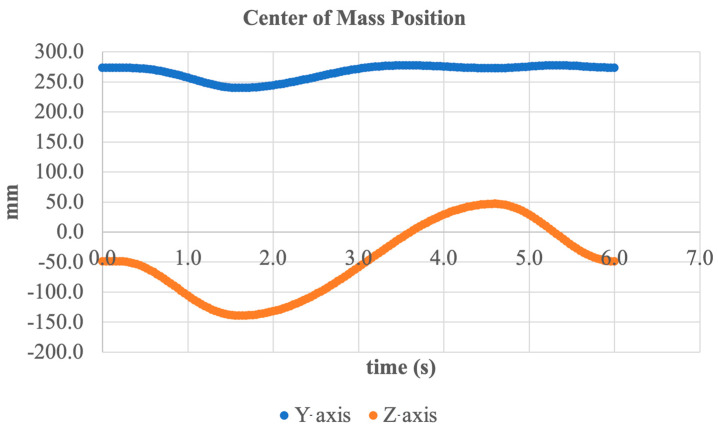
Center of mass position along the Y- and Z-axes during sagittal plane motion.

**Figure 25 sensors-25-04881-f025:**
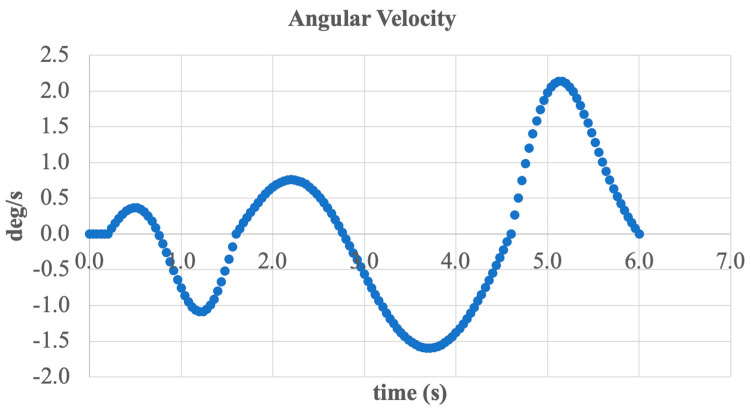
Angular velocity of the foot during dorsiflexion and plantarflexion.

**Figure 26 sensors-25-04881-f026:**
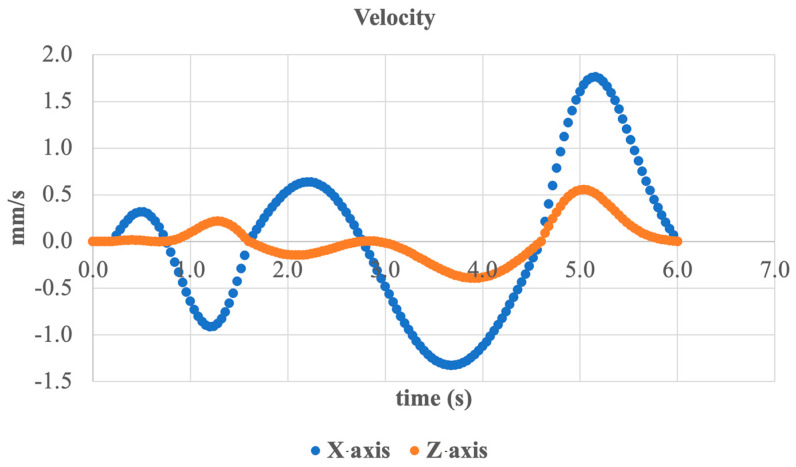
Linear velocity of the foot along the X- and Z-axes.

**Figure 27 sensors-25-04881-f027:**
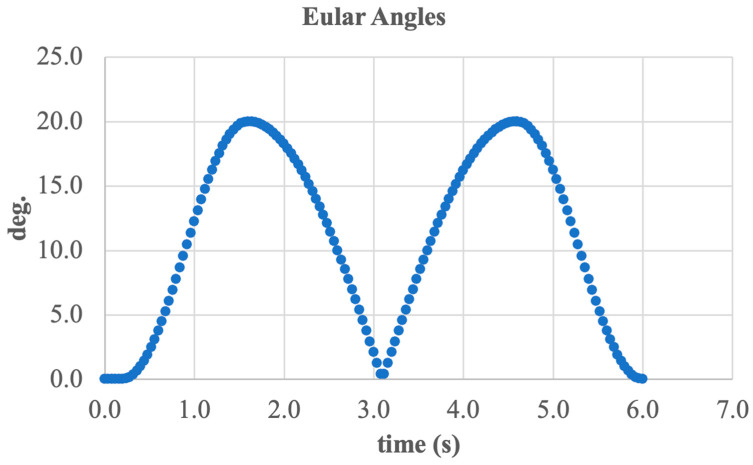
Variation in Euler angles during dorsiflexion/plantarflexion motion.

**Figure 28 sensors-25-04881-f028:**
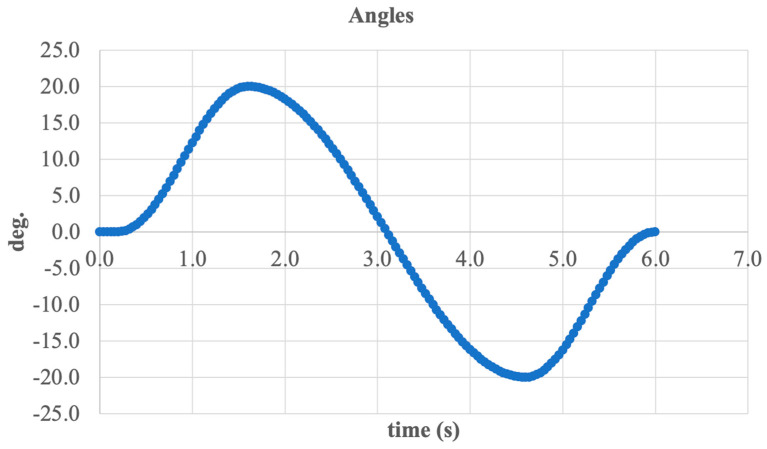
Total angular displacement of the foot over one gait cycle.

**Figure 29 sensors-25-04881-f029:**
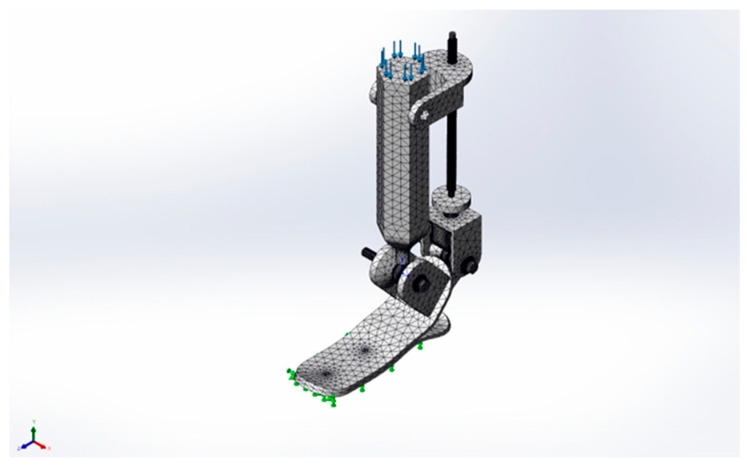
Meshed finite element model of the 2-DOF ankle joint mechanism prepared for FEA.

**Figure 30 sensors-25-04881-f030:**
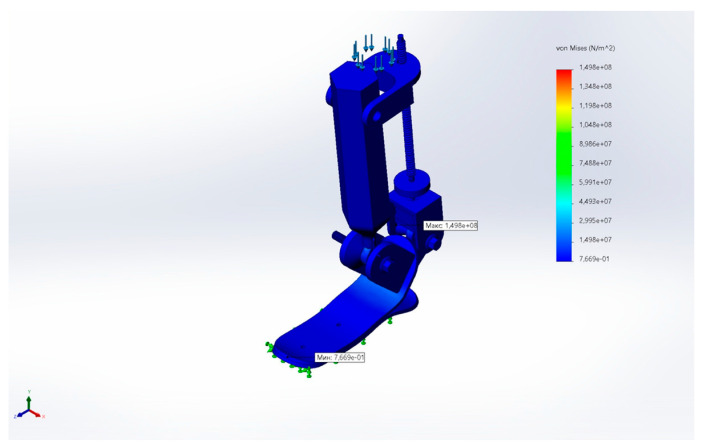
von Mises stress distribution under the applied loading.

**Figure 31 sensors-25-04881-f031:**
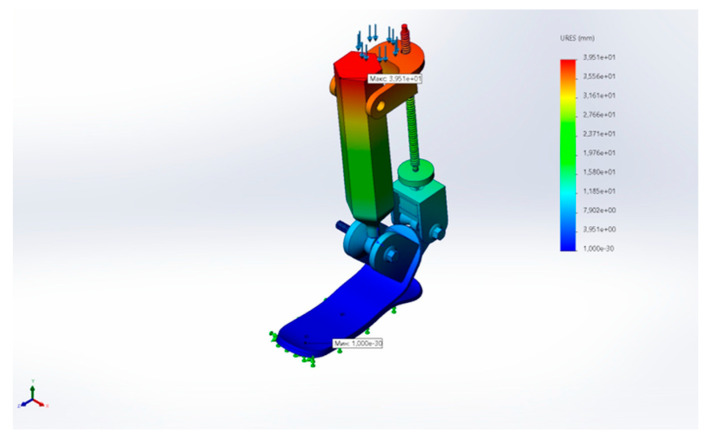
Total displacement (URES) of the mechanism under load.

**Figure 32 sensors-25-04881-f032:**
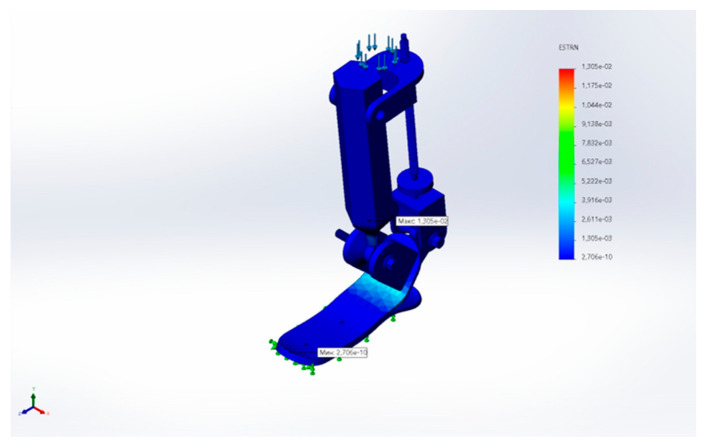
Equivalent strain (ESTRN) distribution under load.

**Table 1 sensors-25-04881-t001:** Quantitative comparison of existing ankle–foot prosthesis designs (Studies [5,6,7,8,9,10,11,12,13,14,15,16,17,18,19]).

Study	DoF	Mass [kg]	Torque [Nm]	ROM [deg]	Power Draw [W]
[5]	2	N/A	N/A	±15	High (external)
[6]	1	N/A	N/A	±10	Passive
[7]	1	1.93	130	±20	Battery
[8]	1	2.2	120	±25	Low
[9]	2	Light	N/A	±18	Low
[10]	2	3.1	90	±18	Moderate
[11]	1	N/A	N/A	±16	Moderate
[12]	1	N/A	N/A	±17	High
[13]	1	N/A	N/A	±15	Moderate
[14]	2	3.7	100	±22	Low
[15]	1	N/A	N/A	±12	Compressed Air
[16]	1	1.9	N/A	±10	None
[17]	4	N/A	N/A	±30	High
[18]	1	3.2	N/A	±14	High
[19]	2	3.4	80	±20	High

**Table 2 sensors-25-04881-t002:** Biomechanical characteristics of the ankle joint and requirements for an active prosthesis.

Parameter	Biological Joint	Prosthesis Requirements
Dorsiflexion	~20°	≥20°, with adaptive modulation
Plantarflexion	25–30°	≥25–30°, with controlled dynamics
Inversion / Eversion	±10–15°	±10–15°, with lateral shift compensation
Torque	100–150 N/m	≥120 N/m, with adaptive regulation
Mechanical Work	Up to 0.2 J/kg per step	Comparable level with high energy efficiency
Stability	Surface adaptation	Dynamic stabilization with feedback
Shock Absorption	Impact attenuation	Active and passive damping
Gait Synchronization	Automatic	Intelligent, with phase prediction
Response to External Forces	Immediate adaptation	Adaptive control with predictive capabilities

**Table 3 sensors-25-04881-t003:** Key overall dimensions of the active ankle prosthesis.

Parameter	Value (mm)
Total height	420.7
Foot length	295.2
Foot width	80
Platform height to foot	296.1
Upper mounting length	148.3

## Data Availability

The data presented in this study is available on request from the corresponding author.

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
