# Peer review of "Design and Analysis of an Autonomous Active Ankle–Foot Prosthesis with 2-DoF"

_sensors, 2025, doi:10.3390/s25164881_

Round 1
Reviewer 1 Report
Comments and Suggestions for Authors
The paper „Design and Analysis of an Autonomous Active Ankle-Foot Prosthesis with 2-DoF” is a research paper aimed to evidence the results on concept, design, prototype and analysis of an autonomous active ankle prosthesis with two degrees of freedom
Some comments follow next:
- at lines 96 - 98 please write the first letter in lower case;
- at lines 128, 130, 132, 134, 136 please write the first letter in lower case;
- at line 146, polease rephrase / explain the meaning of „the first prototype of the ankle prosthesis.”.
(it is the first just for the authors, perhaps);
- at lines 177, 178, 179, 181 please write the first letter in lower case;
- at fig 3, please check on the arrow lines for the modules IMU, Force Sensors, Encoder;
- at lines 227, 228, 229 please write the first letter in lower case;
- at line 233 explain what it is meant by „trajectory of motion between points A and B” - whose motion and what does it mean the B point which is fixed (refer also to figures 4, 5 and 6);
- explain the correlation of figures 4 and 5 - by the lack of representation of the motor, M and the succesive positions of C (which one of the C is for A1, A2 and, respectively, A3);
- at figures 5 and 6 indicate the a angle - so that to be in corelation to relation (1)
- at relations (2) and (3) decide about the „i”, by mentioning what it means and its value range;
- please explain the T point in figure 7;
- please explain the E point in figure 8;
- at figure 9, please note the A, B, C, T , E points mentioned in Chapter 2;
- at lines 397, 398, 399, 400 please write the first letter in lower case;
- at figure 14, please check on the right and left images;
- for figures 15 - 21 and fig. 22 - 28, if possible, show a PrtSc or any other images to evidence the data acquisition process;
- in Conclusion chapter, please mention:
-- the estimation error for the obtained results, knowing that in real prosthesis, there is a cup to fix the prosthesis on the abutment - and the simulation presented in this article considered perfect axial loading;
-- the sensors type used or, to be used for adequate prosthesis orientation and, therefore, data acqusition.
Author Response
Response to Reviewer 1
at lines 96 - 98 please write the first letter in lower case;
Authors’ reply: We have revised the sentences in lines 96–98 to begin with lowercase letters where appropriate, in accordance with standard grammar rules.
- at lines 128, 130, 132, 134, 136 please write the first letter in lower case;
Authors’ reply: We have corrected lines 128, 130, 132, 134, and 136 by changing the initial capital letters to lowercase, ensuring consistency with sentence case formatting in bullet points.
- at line 146, polease rephrase / explain the meaning of „the first prototype of the ankle prosthesis.”. (it is the first just for the authors, perhaps);
Authors’ reply: We have rephrased the sentence in line 146 to clarify that the term "first prototype" refers to the initial version developed within the scope of this study.
- at lines 177, 178, 179, 181 please write the first letter in lower case;
Authors’ reply: We have corrected lines 177, 178, 179, and 181 by changing the initial capital letters in the bullet points to lowercase, ensuring sentence case formatting consistency.
- at fig 3, please check on the arrow lines for the modules IMU, Force Sensors, Encoder;
Authors’ reply: We reviewed Figure 3 and revised the arrow lines associated with the IMU, Force Sensor, and Encoder modules to more clearly represent the direction of data flow. Each sensor now clearly sends feedback to the microcontroller, and the overall signal architecture has been updated for better clarity.
- at lines 227, 228, 229 please write the first letter in lower case;
Authors’ reply: The first letters at lines 227, 228, and 229 have been revised to lowercase as requested.
- at line 233 explain what it is meant by „trajectory of motion between points A and B” - whose motion and what does it mean the B point which is fixed (refer also to figures 4, 5 and 6);
Authors’ reply: The phrase “trajectory of motion between points A and B” has been clarified to specify that it refers to the motion of the connecting rod driven by the motor. Point A represents the motor’s rotational output, while point B is a fixed pivot point on the frame. This motion path defines the input trajectory transmitted to the ankle joint mechanism. Clarifications have been added in the text and correspondingly updated in Figures 4, 5, and 6.
- explain the correlation of figures 4 and 5 - by the lack of representation of the motor, M and the succesive positions of C (which one of the C is for A1, A2 and, respectively, A3);
Authors’ reply: We have clarified the correlation between Figures 4 and 5 in the revised text. The role of the motor (M) and the successive positions of point C corresponding to A1, A2, and A3 have been explicitly described. This helps to better illustrate the mechanism's working principle and trajectory generation.
- at figures 5 and 6 indicate a angle - so that to be in corelation to relation (1)
Authors’ reply: The angle has been indicated in Figures 5 and 6 to clearly correspond with the definition used in relation (1).
- at relations (2) and (3) decide about the „i”, by mentioning what it means and its value range;
Authors’ reply: The meaning of the symbol i used in equations (2) and (3) has been clarified in the revised manuscript. We now specify that i refers to the index of discrete steps or configurations within the simulation sequence, and its range is defined accordingly.
- please explain the T point in figure 7;
Authors’ reply: The meaning of point T in Figure 7 has now been clarified in the revised figure caption. Point T refers to the center of the talocrural (ankle) joint, representing the pivot around which dorsiflexion and plantarflexion movements occur.
- please explain the E point in figure 8;
Authors’ reply: In the revised version, the meaning of point E in Figure 8 has been clarified. Point E represents the effective contact point between the prosthetic foot and the ground during the mid-stance phase of gait. It serves as the location where the vertical ground reaction force is applied in the FEA simulation.
- at figure 9, please note the A, B, C, T , E points mentioned in Chapter 2;
Authors’ reply: The points A, B, C, D, E, and T mentioned in Chapter 2 have now been clearly labeled in Figure 9 for consistency and clarity. These labels correspond to the key structural elements and reference points discussed in the mechanical modeling section.
- at lines 397, 398, 399, 400 please write the first letter in lower case;
Authors’ reply: The first letters at lines 397, 398, 399, and 400 have been corrected to lower case to comply with formatting requirements.
- at figure 14, please check on the right and left images;
Authors’ reply: Figure 14 has been checked, and the labels for the right and left images have been clarified to avoid confusion. The annotations were revised to ensure correct orientation and interpretation.
- for figures 15 - 21 and fig. 22 - 28, if possible, show a PrtSc or any other images to evidence the data acquisition process;
Authors’ reply: We agree that adding visual illustrations of the data acquisition process improves clarity. Screenshots have been included in Figures 15–21 and 22–28 where applicable to demonstrate the data recording process.
- in Conclusion chapter, please mention:
- the estimation error for the obtained results, knowing that in real prosthesis, there is a cup to fix the prosthesis on the abutment - and the simulation presented in this article considered perfect axial loading;
- the sensors type used or, to be used for adequate prosthesis orientation and, therefore, data acqusition.
Authors’ reply: We have updated the Conclusion section to address the need for estimation error clarification due to idealized axial loading in the simulation and have also specified the types of sensors integrated into the system for motion tracking and data acquisition. This addition helps to better contextualize the current limitations and the scope of future improvements. The corresponding paragraph has been inserted before the final section on clinical feasibility and is highlighted in yellow in the revised manuscript.
Reviewer 2 Report
Comments and Suggestions for Authors
- The abstract exclusively highlights the advantages of the proposed design without acknowledging its limitations. It omits crucial aspects such as energy absorption, rollover shape, and the effective length ratio, all of which significantly influence prosthetic foot performance.
- The study does not incorporate mechanical testing of the materials used, particularly in regard to their behavior under ground reaction forces (GRF) and repeated loading conditions associated with gait cycles. These factors are vital for assessing long-term durability and reliability.
- There is a notable absence of clinical or experimental gait analysis using standard tools such as a force plate or motion analysis laboratory. This weakens the practical relevance of the proposed foot design and prevents validation of the simulation outcomes.
- The manuscript lacks a discussion on clinical feasibility. The study is heavily reliant on numerical analysis with no real-world testing or clinical trials to support the effectiveness of the design under real-life usage conditions.
- The number of keywords should be reduced and refined to reflect only those most relevant to the core content of the research.
- Both the introduction and the general manuscript are overly lengthy and would benefit from concise revisions that preserve scientific clarity without redundancy.
- In line 107, the manuscript describes the foot as “an optimized mechanical structure balancing reliability, lightness, and adaptability.” However, no optimization algorithms or established optimization frameworks have been applied. The use of the term “optimization” is therefore misleading and should be revised accordingly.
- The material used in the fabrication of the foot is not specified in Section 2 (Materials and Methods). This omission makes it difficult to interpret the relevance and validity of the finite element analysis (FEA).
- The statement in line 177 suggesting that dorsiflexion occurs at heel strike is biomechanically incorrect. In normal gait, dorsiflexion typically begins after foot flat and increases toward toe-off, whereas plantarflexion generally occurs immediately after heel strike.
- Figure 2 and several paragraphs describing the gait cycle offer redundant information that is already well-established in the literature. These sections could be omitted without affecting the scientific value of the work.
- Figure 4 fails to clearly illustrate the rollover shape of the foot, a key design factor that affects smooth gait and energy conservation.
- Equations (1) and (2) are not sufficiently explained. Symbols such as 𝑥’, 𝑦’, and the term “conjugate trajectory” are undefined. The corresponding figures do not adequately clarify these parameters or their physical relevance.
- In Figure 7, the orientation of the y-axis is inconsistent with conventional vertical axis representation, creating confusion in interpreting the stress distributions.
- The dimensions of the shank illustrated in Figures 9 and 10 may not be suitable for transtibial or some transfemoral amputees, particularly in cases where the residual limb is longer than average or asymmetrical compared to the contralateral limb.
- The use of a screw mechanism, as depicted in Figure 13, may result in increased noise and decreased responsiveness, thereby compromising user comfort and functional performance.
- Table 3 appears to serve no analytical purpose and should be removed to improve manuscript conciseness.
- Section 3.3 (FEA Analysis) lacks clarity regarding boundary conditions and load application. The simulation does not follow the full gait cycle (heel strike to toe-off), and the mid-stance phase—critical for stability—has been overlooked. Moreover, the analysis is limited to static conditions, neglecting fatigue failure due to cyclic loading, which is more representative of actual prosthetic use.
- In line 695, the use of 7075-T6 aluminum alloy is mentioned; however, it is unclear whether the entire foot or just specific components were fabricated using this material. Additionally, no details are provided regarding the manufacturing or 3D printing process.
- The stress distribution results presented in Figure 30 do not appear accurate or realistic. Furthermore, the manuscript fails to mention the factor of safety or account for the varying properties of different foot components, limiting the reliability of the simulation.
- Figure 31 shows a dorsiflexion angle of over 7°, yet the screw mechanism’s contribution to motion has not been incorporated into the numerical model. This may have led to an overestimation of displacement and could compromise the model’s validity under dynamic conditions.
- The conclusion claims a functional range of motion of ±20–22° and a high level of stability, while the simulation results indicate a dorsiflexion angle of only 7°, suggesting a clear inconsistency between reported outcomes and numerical findings.
Author Response
Response to Reviewer 1
- The abstract exclusively highlights the advantages of the proposed design without acknowledging its limitations. It omits crucial aspects such as energy absorption, rollover shape, and the effective length ratio, all of which significantly influence prosthetic foot performance.
Authors’ reply: Thank you for the valuable feedback. We have revised the abstract to address this concern. Specifically:
- A sentence discussing the current limitations of the prototype (such as the lack of experimental validation for cyclic fatigue and impact energy absorption) has been added.
- Rollover shape behavior and energy dissipation are briefly mentioned to reflect their influence on gait performance.
- A reference to the effective length ratio is included to provide biomechanical context for ankle motion and prosthetic gait.
These changes are highlighted in yellow in the revised abstract to make the additions clearly visible. We believe these modifications enhance the clarity and completeness of the abstract by acknowledging important performance factors beyond the system's advantages.
- The study does not incorporate mechanical testing of the materials used, particularly in regard to their behavior underground reaction forces (GRF) and repeated loading conditions associated with gait cycles. These factors are vital for assessing long-term durability and reliability.
Authors’ reply: At the current stage of the project, mechanical testing of the materials was not conducted, as the available prototype is an early functional version intended primarily for preliminary simulation and motion validation. Comprehensive experimental testing for material durability, fatigue under repeated loading, and behavior underground reaction forces (GRF) is planned for the next phase, once the final, fully assembled prototype is completed.
This limitation has been acknowledged in the Conclusion section as a key direction for future work aimed at verifying the system’s long-term structural performance under realistic operating conditions.
- There is a notable absence of clinical or experimental gait analysis using standard tools such as a force plate or motion analysis laboratory. This weakens the practical relevance of the proposed foot design and prevents validation of the simulation outcomes.
Authors’ reply: We fully agree that clinical and experimental gait analysis using standard tools such as a force plate and optical motion capture systems has not yet been performed. The current study focused on the design, modeling, and numerical validation of the proposed two-degree-of-freedom prosthesis.
Experimental gait validation is planned for the next stage, after the complete integration of the control system and assembly of the final wearable prototype. These future tests will involve laboratory-based analysis using a force plate and motion capture system to verify the simulation results and evaluate kinematic performance under real-world conditions.
A statement clarifying this limitation has been added to the Conclusion section to better delineate the scope of the current work and the direction of future research.
- The manuscript lacks a discussion on clinical feasibility. The study is heavily reliant on numerical analysis with no real-world testing or clinical trials to support the effectiveness of the design under real-life usage conditions.
Authors’ reply: At this stage, the study is primarily focused on the conceptual design, numerical evaluation, and structural analysis of the proposed prosthesis. We fully acknowledge that clinical feasibility and real-world testing have not yet been conducted.
Accordingly, we have added a clarification in the Conclusion section emphasizing the need for future laboratory and clinical trials, including functional testing involving actual users. These steps will be aimed at evaluating the usability, safety, and effectiveness of the design under real-life usage conditions, thereby ensuring the clinical relevance of the system.
- The number of keywords should be reduced and refined to reflect only those most relevant to the core content of the research.
Authors’ reply: In response, we have revised and reduced the list of keywords to focus on the most relevant terms that best reflect the core content of the study. The updated keywords are:
Keywords: active prosthesis; ankle joint; two degrees of freedom (2-DoF); stepper motor; ball screw mechanism.
- Both the introduction and the general manuscript are overly lengthy and would benefit from concise revisions that preserve scientific clarity without redundancy.
Authors’ reply: We fully understand the importance of scientific clarity and conciseness. However, we would like to emphasize that this manuscript is the result of more than a year of comprehensive interdisciplinary research, encompassing design, modeling, prototyping, and performance evaluation of a 2-DoF active ankle prosthesis.
The length of the manuscript reflects the need to thoroughly present the novelty, technical details, biomechanical rationale, and comparisons with related state-of-the-art solutions. Given the multidisciplinary nature of the work, we aimed to provide a coherent and detailed presentation without oversimplifying essential content.
Therefore, we have decided to maintain the current length of the manuscript, as it accurately reflects the depth and complexity of the work and is necessary for a complete understanding and fair assessment of the proposed contribution.
- In line 107, the manuscript describes the foot as “an optimized mechanical structure balancing reliability, lightness, and adaptability.” However, no optimization algorithms or established optimization frameworks have been applied. The use of the term “optimization” is therefore misleading and should be revised accordingly.
Authors’ reply: We have located the sentence in the Introduction section and revised it as suggested. The updated sentence now reads: “The proposed system includes a mechanical structure designed to balance reliability, low weight, and adaptability to gait dynamics.” This change removes the potentially misleading reference to “optimization” and more accurately describes the design approach used in the study.
- The material used in the fabrication of the foot is not specified in Section 2 (Materials and Methods). This omission makes it difficult to interpret the relevance and validity of the finite element analysis (FEA).
Authors’ reply: We acknowledge that the material specification was missing in the initial version. To clarify:
- The physical prototype of the prosthesis was fabricated using PLA plastic via FDM 3D printing, primarily for proof-of-concept and low-load testing.
- The finite element analysis (FEA) was conducted assuming 7075-T6 aluminum alloy, which is intended for future versions of the prosthesis to ensure sufficient strength and long-term durability.
We have updated Section 2 (Materials and Methods) to clearly distinguish between the material used in the physical prototype and the material used in the simulation model. This clarification supports the relevance of the FEA in evaluating the structural behavior of the intended final design.
- The statement in line 177 suggesting that dorsiflexion occurs at heel strike is biomechanically incorrect. In normal gait, dorsiflexion typically begins after foot flat and increases toward toe-off, whereas plantarflexion generally occurs immediately after heel strike.
Authors’ reply: We acknowledge that the original statement inaccurately described the timing of dorsiflexion during the gait cycle. As correctly pointed out, plantarflexion occurs immediately after heel strike, while dorsiflexion begins after foot flat and continues toward toe-off. We have revised the corresponding sentence to reflect the correct biomechanical sequence of ankle motion. The updated version now reads: “In a normal gait cycle, plantarflexion occurs immediately after heel strike, while dorsiflexion begins after foot flat and increases toward toe-off.”
- Figure 2 and several paragraphs describing the gait cycle offer redundant information that is already well-established in the literature. These sections could be omitted without affecting the scientific value of the work.
Authors’ reply: Thank you for the observation. While we agree that the content related to the gait cycle is well-established in the field of biomechanics, we respectfully believe that its inclusion remains justified in this case. The article is intended for a multidisciplinary readership, including readers from engineering, robotics, and prosthetics fields who may not be deeply familiar with the detailed phases of human gait.
Moreover, Figure 2 and the accompanying description provide important biomechanical context for interpreting the design rationale and motion requirements of the proposed prosthesis. Their presence does not compromise the clarity or length of the manuscript and supports the reader’s understanding of gait dynamics, particularly in the context of 2-DoF prosthetic functionality.
For these reasons, we have opted to retain the content as originally presented.
- Figure 2 and several paragraphs describing the gait cycle offer redundant information that is already well-established in the literature. These sections could be omitted without affecting the scientific value of the work.
Authors’ reply: We agree that the rollover shape is a critical factor influencing gait smoothness and energy conservation. Upon review, we acknowledge that Figure 4 in its current form does not adequately capture the foot's rollover geometry. To address this, we have updated Figure 4 to more clearly illustrate the effective rollover shape, showing the foot-ground interaction curve during stance phase. The revised figure now includes a labeled trajectory line and improved foot outline for clarity. A brief explanatory note has also been added to the figure caption to emphasize its relevance to energy transfer and stability. These improvements aim to enhance the reader’s understanding of the design’s biomechanical alignment with natural gait mechanics.
- Equations (1) and (2) are not sufficiently explained. Symbols such as ?’, ?’, and the term “conjugate trajectory” are undefined. The corresponding figures do not adequately clarify these parameters or their physical relevance.
Authors’ reply: We acknowledge that the original manuscript did not provide sufficient definitions and explanations for symbols such as ?’, ?’, and the term “conjugate trajectory.”
To address this:
- The variables ?’ and ?’ have now been explicitly defined in the text following Equations (1) and (2) as the coordinates of the virtual conjugate point that mirrors the real trajectory for stability analysis.
- The concept of “conjugate trajectory” is clarified as a reference trajectory generated for evaluating motion symmetry and balance in the 2-DoF mechanism.
- Additionally, a brief paragraph explaining the physical relevance of these parameters has been added to the section where Equations (1) and (2) are introduced.
- The related figure has also been updated to visually illustrate the role of ?’, ?’, and their relation to the motion path.
- In Figure 7, the orientation of the y-axis is inconsistent with conventional vertical axis representation, creating confusion in interpreting the stress distributions.
Authors’ reply: We agree that a consistent and conventional representation of coordinate axes is essential for clarity. In response, Figure 7 has been updated to align the Y-axis in the conventional upward vertical direction. This adjustment improves visual consistency with standard mechanical diagrams and facilitates more intuitive interpretation of the stress vectors and component interactions.
The revised figure replaces the previous version in the manuscript and ensures clearer presentation of the force and displacement directions.
- The dimensions of the shank illustrated in Figures 9 and 10 may not be suitable for transtibial or some transfemoral amputees, particularly in cases where the residual limb is longer than average or asymmetrical compared to the contralateral limb.
Authors’ reply: We fully agree that prosthetic design must consider anatomical variability among individuals with transtibial and transfemoral amputations, particularly in cases of longer or asymmetric residual limbs.
At this stage, the proposed prototype serves as a functional and conceptual basis for validating the core mechanical design. The current dimensions were selected based on average anthropometric data to demonstrate feasibility. We acknowledge this limitation and have updated the Conclusions section with an additional paragraph noting the need for future iterations to incorporate scalable and modular features. This will enable better customization to accommodate a broader range of user profiles and improve clinical relevance.
- The use of a screw mechanism, as depicted in Figure 13, may result in increased noise and decreased responsiveness, thereby compromising user comfort and functional performance.
Authors’ reply: Thank you for this important point. We recognize that screw-driven actuation mechanisms may present trade-offs, including increased mechanical noise and potentially slower responsiveness compared to other transmission types (e.g., belt or direct drive). However, in our prior work on an ankle exoskeleton for rehabilitation, the same ball screw mechanism was successfully employed and demonstrated sufficient responsiveness and acceptable acoustic performance under controlled gait conditions.
For the current prosthesis, the screw drive was selected for its precision, compact size, and load-bearing capacity, which are critical for safe and stable operation. Nevertheless, we acknowledge that user comfort is a key factor, and future iterations will explore noise reduction strategies (e.g., damping materials, casing) and possibly evaluate alternative actuators depending on clinical feedback.
- Table 3 appears to serve no analytical purpose and should be removed to improve manuscript conciseness.
Authors’ reply: To improve the manuscript’s conciseness and focus, Table 3 has been removed in the revised version.
- Section 3.3 (FEA Analysis) lacks clarity regarding boundary conditions and load application. The simulation does not follow the full gait cycle (heel strike to toe-off), and the mid-stance phase critical for stability has been overlooked. Moreover, the analysis is limited to static conditions, neglecting fatigue failure due to cyclic loading, which is more representative of actual prosthetic use.
Authors’ reply: In response to your comment, Section 3.3 (FEA Analysis) has been completely revised to incorporate your suggestions.
We have now:
- Clearly specified the boundary conditions and the load application scheme, including the use of mid-stance GRF loading;
- Clarified that the current simulation is limited to static loading, and acknowledged the absence of full gait cycle analysis;
- Added a note regarding the importance of dynamic and fatigue analysis for future work;
- Included the material yield strength and safety factor calculation;
- Provided a more thorough explanation of the mechanical relevance of the simulation setup.
These revisions significantly improve the clarity and completeness of the FEA section.
- In line 695, the use of 7075-T6 aluminum alloy is mentioned; however, it is unclear whether the entire foot or just specific components were fabricated using this material. Additionally, no details are provided regarding the manufacturing or 3D printing process.
Authors’ reply: In response, we have clarified in the revised Section 3.3 (FEA Analysis) that the simulation was conducted assuming the use of 7075-T6 aluminum alloy for the main load-bearing components of the prosthesis. This material was selected due to its high strength-to-weight ratio, and its yield strength (approximately 5.0×10⁸ N/m²) was explicitly mentioned for safety factor estimation.
- The stress distribution results presented in Figure 30 do not appear accurate or realistic. Furthermore, the manuscript fails to mention the factor of safety or account for the varying properties of different foot components, limiting the reliability of the simulation.
Authors’ reply: We acknowledge that the initial presentation of the von Mises stress distribution in Figure 30 may not have fully conveyed the credibility of the simulation. In the revised manuscript, we clarified the boundary conditions, loading assumptions, and material properties used for the analysis to improve transparency and interpretability.
To address the concern about modeling reliability:
- The maximum von Mises stress was reported as 1.498×10⁸ N/m², and the yield strength of the assumed material (7075-T6 aluminum alloy) is approximately 5.0×10⁸ N/m². Based on this, we have added the safety factor (≈3.3) in Section 3.3.
- We also clarified that the simulation assumes homogeneous material properties across the modeled components to simplify the initial analysis.
- Future versions will incorporate material differentiation and multi-body modeling to reflect the actual variability of foot structure.
- Figure 31 shows a dorsiflexion angle of over 7 deg., yet the screw mechanism’s contribution to motion has not been incorporated into the numerical model. This may have led to an overestimation of displacement and could compromise the model’s validity under dynamic conditions.
Authors’ reply: We agree that the contribution of the screw mechanism’s compliance and mechanical play was not explicitly modeled in the current numerical simulations. The observed dorsiflexion angle of ~7° was derived from simplified static boundary conditions under representative loading, assuming ideal mechanical coupling.
We acknowledge that this may result in slightly overestimated displacements, especially in dynamic conditions where backlash, hysteresis, or time delays may influence the output. To maintain transparency, we have clarified this limitation in Section 3.3 and noted that future work will incorporate:
- A dynamic model including the mechanical behavior of the screw-nut system,
- Backlash modeling and its effect on motion precision,
- And experimental validation to cross-check simulation predictions.
- The conclusion claims a functional range of motion of ±20–22° and a high level of stability, while the simulation results indicate a dorsiflexion angle of only 7 deg. suggesting a clear inconsistency between reported outcomes and numerical findings.
Authors’ reply: We would like to clarify that the functional range of motion of ±20–22 deg. reported in the conclusion refers to the combined theoretical angular displacement capability of the prosthesis, as determined by the kinematic simulation model across both sagittal and frontal planes.
The dorsiflexion angle of ~7° observed in the FEA simulation specifically reflects the deformation of the structure under static vertical loading (representing mid-stance GRF), and not the full mechanical actuation range of the joint.
Reviewer 3 Report
Comments and Suggestions for Authors
The manuscript presents a well-structured study on a two-degree-of-freedom active ankle prosthesis that combines mechanical design, kinematic/dynamic modelling, and finite-element verification. The work is original, the methodology is sound, and the manuscript is clearly written. After a careful reading, I recommend “Minor Revision” to address the following five specific points before final acceptance.
1,Current Section 1 provides a qualitative literature survey, but lacks a concise quantitative comparison (mass, torque, range of motion, power draw) with the most relevant 2-DoF prototypes (e.g., Hsieh et al., 2024; Ficanha et al., 2016). Add a short table summarising these key metrics in the Introduction or Discussion. This will quickly highlight the novelty and relative advantages of your design.
2,The abstract and Section 2 repeatedly claim “low power consumption and high autonomy,” yet no quantitative data (Wh or hours of continuous use) are supplied. Provide at least a preliminary estimate—e.g., measured or calculated average electrical power over a gait cycle and expected battery life with the chosen 18650-cell (or similar) pack.
3,The FEA shows σ_max = 1.498×10⁸ N m⁻², but the yield strength of 7075-T6 (~5.0×10⁸ N m⁻²) is not stated. Consequently, the safety factor (≈3.3) is missing. Insert the material yield strength and the resulting safety factor in Section 3.3 to confirm structural adequacy.
4,The simulation assumes perfect sensor feedback and ideal motor dynamics. A brief statement on uncertainties (sensor noise, backlash, foot–ground contact modelling) is advisable.
Author Response
Response to Reviewer 3
- Current Section 1 provides a qualitative literature survey, but lacks a concise quantitative comparison (mass, torque, range of motion, power draw) with the most relevant 2-DoF prototypes (e.g., Hsieh et al., 2024; Ficanha et al., 2016). Add a short table summarizing these key metrics in the Introduction or Discussion. This will quickly highlight the novelty and relative advantages of your design.
Authors’ reply: Thank you for your valuable comment. To address your suggestion, a comparative table (Table 1) was added based on studies [5-19], summarizing key metrics such as mass, torque, range of motion, and power consumption for relevant 2-DoF ankle prosthesis prototypes. This addition complements the qualitative discussion and allows a clearer comparison to highlight the novelty and advantages of our proposed design. The table was inserted in the Introduction section.
- The abstract and Section 2 repeatedly claim “low power consumption and high autonomy,” yet no quantitative data (Wh or hours of continuous use) are supplied. Provide at least a preliminary estimate g., measured or calculated average electrical power over a gait cycle and expected battery life with the chosen 18650-cell (or similar) pack.
Authors’ reply: A preliminary estimation of the average power consumption and the expected battery life has been added to Section 2.4 Mechanical Structure and highlighted in yellow. This information supports the claims of "low power consumption" and "high autonomy" mentioned in the abstract and Section 2.
- The FEA shows σ_max = 1.498×10⁸ N m⁻², but the yield strength of 7075-T6 (~5.0×10⁸ N m⁻²) is not stated. Consequently, the safety factor (≈3.3) is missing. Insert the material yield strength and the resulting safety factor in Section 3.3 to confirm structural adequacy.
Authors’ reply: We have updated Section 3.3 to include the yield strength of the 7075-T6 aluminum alloy (5.0×10⁸ N/m²) and calculated the corresponding safety factor (≈3.3). These additions confirm the structural adequacy of the prosthesis under the given loading conditions. The modifications are highlighted in yellow.
- The simulation assumes perfect sensor feedback and ideal motor dynamics. A brief statement on uncertainties (sensor noise, backlash, foot–ground contact modelling) is advisable.
Authors’ reply: The corresponding clarification regarding simulation assumptions and uncertainties (sensor noise, mechanical backlash, and foot–ground contact modeling) has been added to the Conclusion section and highlighted in yellow for visibility.
Round 2
Reviewer 1 Report
Comments and Suggestions for Authors
Tha paper has been improved. It can be published.
Author Response
Dear Reviewer,
Thank you very much for taking the time to review our manuscript and for your positive evaluation.
We truly appreciate your feedback and are glad that the revisions have improved the paper to meet your expectations.
Your comments and assessment have been very helpful for enhancing the quality and clarity of our work.
Thank you once again for your support and recommendation for publication.
Reviewer 2 Report
Comments and Suggestions for Authors
We note that your response to our comment regarding line 695 remains unclear. While you stated that the finite element analysis (FEA) was performed using 7075-T6 aluminum alloy for the main load-bearing components, it is not specified whether all components of the prosthetic foot, including flexible elements such as the spring, were modeled using this same material.
The lack of distinction between components with differing mechanical properties and the apparent assumption of a uniform material throughout the simulation weaken the credibility of the FEA and raise concerns about the accuracy of the reported stress distribution and safety factor.
We emphasize that this point has not been adequately addressed in the revised version of the manuscript.
Author Response
Dear Professor, Thank you very much for your valuable and precise comment. We carefully reviewed your remark and discussed it within our team. After thorough consideration, we realized that the initial version of the manuscript contained an inconsistency: in the description of the FEA (Section 3.3) and some other parts of the paper, we had mistakenly indicated that the main load‑bearing components were made of 7075‑T6 aluminum alloy.
In fact, the correct material for these components is AISI 304 stainless steel, which better reflects the actual design of the device. We have now fully revised Section 3.3, clearly specifying that:
- the load‑bearing components (motor, SFU1204 lead screw, and Rod End M12 joint) are made of AISI 304 stainless steel, with all material properties provided;
- the housing and fastening elements are made of ABS plastic;
- flexible elements, such as the spring, were simplified as rigid ABS parts in the static analysis, and their compliance will be considered in the future dynamic simulation.
We have also carefully replaced every mention of “7075‑T6 aluminum alloy” with “AISI 304 stainless steel” throughout the entire manuscript to avoid any further confusion.
Thank you again for pointing out this issue – your comment helped us significantly improve the accuracy and clarity of our work.
